# Histone Deacetylase 7 mediates tissue-specific autoimmunity via control of innate effector function in invariant Natural Killer T Cells

Herbert G Kasler[1,2,3†], Intelly S Lee[1,2†], Hyung W Lim[1,2‡], Eric Verdin[1,2,3*]

[1]Gladstone Institute of Virology and Immunology, San Francisco, United States; [2]Department of Medicine, University of California, San Francisco, San Francisco, United States; [3]Buck Institute for Research on Aging, Novato, United States

**Abstract** We report that Histone Deacetylase 7 (HDAC7) controls the thymic effector programming of Natural Killer T (NKT) cells, and that interference with this function contributes to tissue-specific autoimmunity. Gain of HDAC7 function in thymocytes blocks both negative selection and NKT development, and diverts Vα14/Jα18 TCR transgenic thymocytes into a Tconv-like lineage. Conversely, HDAC7 deletion promotes thymocyte apoptosis and causes expansion of innate-effector cells. Investigating the mechanisms involved, we found that HDAC7 binds PLZF and modulates PLZF-dependent transcription. Moreover, HDAC7 and many of its transcriptional targets are human risk loci for IBD and PSC, autoimmune diseases that strikingly resemble the disease we observe in HDAC7 gain-of-function in mice. Importantly, reconstitution of iNKT cells in these mice mitigated their disease, suggesting that the combined defects in negative selection and iNKT cells due to altered HDAC7 function can cause tissue-restricted autoimmunity, a finding that may explain the association between HDAC7 and hepatobiliary autoimmunity.

DOI: https://doi.org/10.7554/eLife.32109.001

*For correspondence:
everdin@buckinstitute.org

†These authors contributed equally to this work

Present address: ‡Novartis Institutes for Biomedical Research (NIBR), Inc, Cambridge, United States

## Introduction

To become mature T cells, thymocytes must navigate through a complex process of selection and instruction, centered around signals received through their newly created T cell antigen receptors (TCRs). For thymocytes destined to become conventional naïve CD4 or CD8 T cells (Tconv), this requires passing two key checkpoints: positive selection, in which cortical CD4/CD8 double-positive (DP) thymocytes must receive a minimum level of TCR stimulation from self peptide-MHC complexes in order to adopt the appropriate lineage and continue maturation, and negative selection, in which thymocytes with self-reactivity above a critical threshold are deleted from the repertoire by activation-induced apoptosis. While the elucidation of these mechanisms decades ago established a basic conceptual framework for the creation of a competent and self-tolerant T cell repertoire, the years since have brought to light an ever-increasing variety of alternate developmental programs that produce specialized populations of mature T cells functionally distinct from Tconv. These populations, critical for both effective host defense and self-tolerance, are elicited from the diverse pool of T cell precursors by specialized selection mechanisms, mostly involving strong recognition of noncanonical ligands, as in the case of NKT cells (*Kronenberg, 2014*), or recognition of peptide-MHC ligands at high TCR avidities near the threshold of negative selection, as in the case of nTreg or CD8αα IEL (*Klein et al., 2014*; *Moran et al., 2011*). The TCR signals involved in their development are generally stronger than those that mediate positive selection to the Tconv lineage, and the process is thus termed agonist selection (*Stritesky et al., 2012*).

**eLife digest** To protect us, our immune system must walk a narrow line: while it eliminates all external threats, it also has to refrain from attacking the healthy tissues of our body. When such misdirected attacks do take place, they can result in life-threatening autoimmune diseases.

T cells are a highly diverse population of immune cells that can recognize and orchestrate the body's response against infected or 'abnormal' cells. Early in the development of most types of T cells, the body normally weeds out the ones that target healthy tissues. A gene known as Histone Deacetylase 7 (*HDAC7*) regulates this process. However, when *HDAC7* carries a specific mutation called *HDAC7-ΔP*, dangerous T cells that can attack healthy tissues 'escape' this selection.

The *HDAC7-ΔP* mutation allows T cells that react to many different tissues to survive. However, in mice with this genetic change, only the liver, the digestive system and the pancreas are actually damaged by the immune system and show signs of autoimmune diseases. Why are these organs affected, and not the others?

Here, Kasler, Lee et al. find that *HDAC7* also helps another type of T cell to develop. Known as invariant natural killer T – or iNKT – cells, these cells specialize in defending the gut, liver and pancreas against bacteria. Mice with the *HDAC7-ΔP* mutation can no longer produce iNKT cells. Remarkably, restoring normal levels of these cells in the *HDAC7-ΔP* animals reduces the symptoms of their autoimmune diseases, even though the mice are still carrying the T cells that have escaped selection and can attack healthy tissues.

Taken together, these results explain why a mutation in *HDAC7* can create problems only for specific organs in the body. However, it is still not clear exactly why losing iNKT cells increases autoimmune attacks of the tissues they normally occupy. One possibility is that these cells limit access to the organs by other immune cells that could cause damage. Another option is that, when iNKT cells are absent, gut bacteria can attack and create an inflammation. This recruits T cells to the site, including the ones that can attack healthy organs.

In humans, mutations in *HDAC7*, as well as in other genes that regulate it, are also associated with autoimmune disorders of the digestive tract and liver. These include inflammatory bowel diseases such as ulcerative colitis or Crohn's disease. Ultimately the findings presented by Kasler, Lee et al. could be a starting point for finding new treatments for these illnesses.

DOI: https://doi.org/10.7554/eLife.32109.002

One feature that distinguishes many of these specialized cell types from Tconv is the thymic acquisition of constitutive effector function, a phenotype shared with innate immune cells and thus giving rise to the term 'innate-like' or 'innate effector' T cells. Whereas Tconv exit the thymus with a naive phenotype, circulate broadly, and require a several days long, orchestrated process of priming and clonal expansion to become fully functional effector/memory cells, innate-like T cells are often constitutively tissue-resident and make mature effector responses to their cognate stimuli immediately (*Kang and Malhotra, 2015*). Innate-like T cells exit the thymus larger than Tconv, with an antigen-experienced phenotype and an expanded secretory apparatus, allowing them to rapidly elaborate robust cytokine responses after brief TCR stimulation (*Kang and Malhotra, 2015*; *Chandra and Kronenberg, 2015*; *Brennan et al., 2013*). These differences arise due an alternative thymic maturation process that parallels the priming of naïve T-cells in the periphery. For NKT cells, this includes a ~ 100 fold intra-thymic proliferative expansion to generate pre-established clonal populations (*Benlagha et al., 2002*). Maintenance of the innate effector phenotype in NKT cells can at least partially be attributed to stable expression of their signature transcription factor Promyelocytic Leukemia Zinc Finger Protein (PLZF, ZBTB16) (*Kovalovsky et al., 2008*; *Savage et al., 2008*). PLZF expression is established during thymic development of NKT cells, via a cellular mechanism that involves strong recognition of glycolipid ligands on the non-canonical CD1D MHC molecule by a clonally restricted (in mice, Vα14/Jα18 with one of several possible β chains) TCR, together with homotypic co-stimulation through the SAP family of co-receptors (*Bendelac et al., 2007*). However, what downstream factors link these surface signals to stable PLZF expression and what other pathways may be involved are still open questions.

We have previously described how Tconv development is regulated by the class IIA histone deacetylase Histone Deacetylase 7 (HDAC7), a TCR signal-regulated corepressor abundantly expressed in thymocytes (*Dequiedt et al., 2003*; *Kasler and Verdin, 2007*). The activity of HDAC7 is controlled by nuclear exclusion in response to phosphorylation of conserved serine residues in their N-terminal adapter domains (*Verdin et al., 2003*). In thymocytes, TCR stimulation results in HDAC7 phosphorylation and nuclear exclusion via Protein Kinase D (*Parra et al., 2005*). CD4/CD8 double-positive (DP) thymocytes lacking HDAC7 are much more likely than WT thymocytes to die before becoming positively selected, significantly impeding their development into mature Tconv (*Kasler et al., 2011*). Conversely, if a transgene encoding a phosphorylation-deficient, constitutively nuclear version of human HDAC7 (HDAC7-ΔP) is transiently expressed in the thymus at sub-endogenous levels (*Kasler et al., 2012*), deletion of autoreactive thymocytes by negative selection is strongly blocked and the hosts develop lethal autoimmunity. Consistent with broad blockade of negative selection, we observed autoantibodies to a comprehensive array of tissue antigens in these mice (*Kasler et al., 2012*). However, for reasons that were not clear to us at the time actual tissue destruction occurred almost exclusively in a gastrointestinal/hepatobiliary compartment that is anatomically tied together by the contiguous epithelial surfaces of the GI lumen and the pancreatic and biliary ductal systems (*Kasler et al., 2012*).

The potential significance of this peculiar pattern of HDAC7-mediated autoimmunity for human disease has recently been brought into sharp focus by two separate studies identifying polymorphisms at the loci of HDAC7 as well as several of its upstream regulatory kinases as independent risk factors in human inflammatory bowel disease (IBD), and also in primary sclerosing cholangitis (PSC), a destructive autoimmune syndrome of the hepatobiliary system, which is additionally associated with increased IBD risk (*Liu et al., 2013*; *Jostins et al., 2012*). The striking parallel between these human syndromes and the autoimmunity observed in HDAC7-ΔP transgenic mice suggested to us a connection between HDAC7 and these types of autoimmunity that goes beyond simply blocking thymic negative selection. This led us to undertake a more thorough phenotypic characterization of mice with altered HDAC7 function during T cell development, revealing that HDAC7 has a key role in the regulation of the innate effector programming of iNKT cells, at least in part via direct modulation of the transcriptional activity of PLZF. Both gain and loss of HDAC7 function in thymocytes resulted in aberrant effector programming of T cells in both the Tconv and innate-like lineages, leading to multiple abnormalities in peripheral populations. These studies shed new light on the molecular pathways that regulate the effector programming of innate-like T cells, reveal a new key molecular target of HDAC7 in T cell development, and set forth a novel cellular model of tissue-specific autoimmunity, in which one genetic lesion mediates multiple defects in thymic selection, which then converge in the periphery to produce a unique, tissue-restricted pattern of disease. Given the established genetic association between *HDAC7* variants and very similar human syndromes, our findings are likely to be of considerable significance in the understanding of these diseases.

## Results

### Alteration of HDAC7 function dysregulates thymic innate effector programming and interferes with iNKT development

We previously showed that if a constitutively nuclear mutant of human HDAC7 (HDAC7-ΔP) is transiently expressed at normal levels during thymic T cell development but not in mature T cells, autoreactive cells that would normally die by negative selection instead exit the thymus as naïve Tconv (*Kasler et al., 2012*). However in our previous study we did not assess the fates of most cells destined to become innate effectors. Analyzing these populations, we noted a modest suppression of Treg (*Kasler et al., 2012*) and CD8αα IEL (*Figure 1—figure supplement 1A*), but the most striking observation we made was the near total absence of invariant Natural Killer T cells (iNKT), an oligoclonal population that is reactive to α-galactosyl ceramide (αGalCer) presented by the CD1D non-canonical MHC molecule (CD1D/αGalCer) (*Kronenberg, 2014*). Cells positive for staining with CD1D/αGalCer tetramers represent approximately 3% of TCRβ-positive cells in wild type C57BL/6 (B6) thymus and 30% in liver, however they are nearly undetectable in either of these tissues or in the spleens or livers of HDAC7-ΔP mice (*Figure 1A,B*; *Figure 1—figure supplement 1B*, for full gating see *Figure 1—figure supplement 1A*), suggesting a profound deficiency in iNKT development.

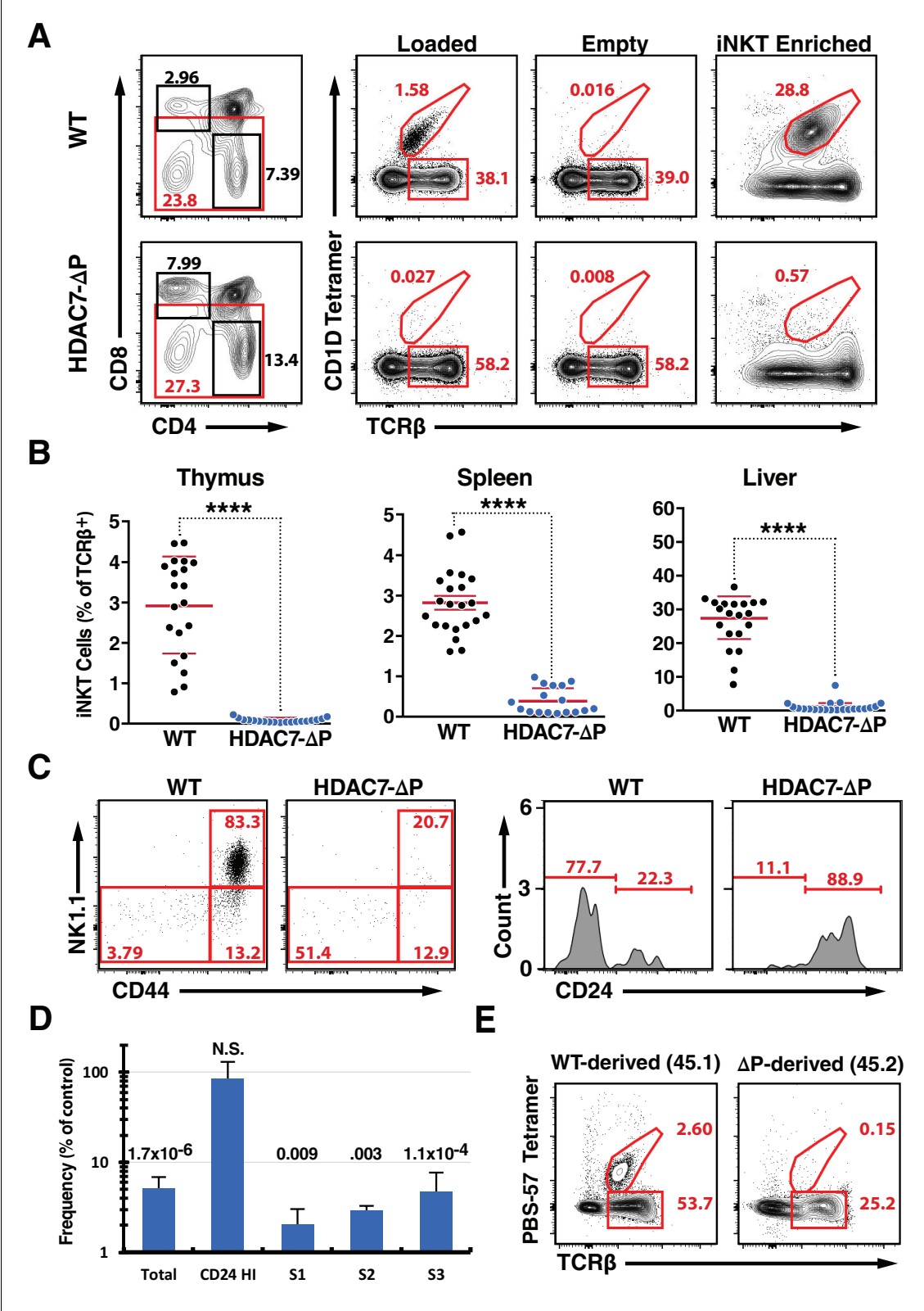

**Figure 1.** A Gain-of-Function HDAC7 Mutant, HDAC7-ΔP, Arrests Thymic iNKT Development. (**A**) Representative flow cytometric plots of iNKT cells and conventional αβ T-cells, identified by staining with TCRβ and CD1D tetramer, empty or loaded with αGalCer as indicated, in thymocytes from WT (top) and HDAC7-ΔP (bottom) mice. Staining after magnetic enrichment of $2 \times 10^7$ cells with loaded tetramer is shown at right. (**B**) Quantification of iNKT cell frequency in Thymus (left), spleen (center), and liver (right) of WT (black symbols) and HDAC7-ΔP (blue symbols) mice (**C**) Representative flow plots

*Figure 1 continued on next page*

Figure 1 continued

showing conventional staging of iNKT development by CD44 and NK1.1 expression (left) and CD24 expression (right) in magnetically enriched Tet⁺ TCRβ⁺ thymic iNKT cells from WT and HDAC7-ΔP mice as indicated. (D) Quantification of difference in frequency of magnetically enriched iNKT cells at the indicated stages, as defined in (C), for five littermate pairs WT and HDAC7-ΔP mice. Difference is expressed as (% of live cells / % of live cells) * 100 for HDAC7-ΔP/WT. Numbers above each column indicate P-value by 2-tailed Student's T-test. (E) Representative flow cytometric plots and of iNKT from thymus in WT (CD45.1): HDAC7-ΔP (CD45.2) mixed bone-marrow chimeras. Data in (B) are combined from eight independent experiments involving 1–3 littermate pairs; data in (C) are representative of 5 WT: HDAC7-ΔP littermate pairs. Data in (E) are representative of 3 sets of chimeras with 3–6 mice per group. Statistical significance was determined using unpaired two-tailed t tests; ****p≤0.0001.
DOI: https://doi.org/10.7554/eLife.32109.003

The following source data and figure supplements are available for figure 1:

Source data 1. Multi-sheet Microsoft Excel workbook containing numerical data matrices for all figure panels (on separate sheets) in which individual data points are not represented graphically.
DOI: https://doi.org/10.7554/eLife.32109.007

Figure supplement 1. Supporting Data on iNKT Phenotype of HDAC7-ΔP Transgenic Mice.
DOI: https://doi.org/10.7554/eLife.32109.004

Figure supplement 2. Supporting Data on the iNKT Phenotype of WT: HDAC7-ΔP Mixed Hematopoietic Chimeras.
DOI: https://doi.org/10.7554/eLife.32109.005

Figure supplement 3. Full Gating Strategy for Detection and Staging of iNKT Cells in HDAC7-ΔP Transgenic Mice and WT: HDAC7-ΔP Mixed Hematopoietic Chimeras.
DOI: https://doi.org/10.7554/eLife.32109.006

There were however consistently more cells detected in the thymus of HDAC7-ΔP transgenic mice with αGalCer-loaded tetramer than with empty tetramer (*Figure 1A*, *Figure 1—figure supplement 1C–D*), suggesting that iNKT cells are not entirely absent in this background.

Analyzing these cells according to the conventional staging system for iNKT development (*Stritesky et al., 2012*), we found that rather than being predominantly CD44^hi/NK1.1^+ (Stage 3), as in the case of WT iNKT cells, the few thymic tetramer-reactive cells from HDAC7-ΔP mice were evenly distributed between the CD44^hi/NK1.1^+, CD44^hi/NK1.1^- (Stage 2), and CD44^lo/NK1.1^- (Stage 0–1) populations (*Figure 1—figure supplement 1D*). Further analysis of the Stage 0–1 population showed these cells to be predominantly CD24^hi, indicating a profound reduction in numbers at all developmental stages that were detectable above background (*Figure 1—figure supplement 1C*). Examining these stages after ~20 fold enrichment of iNKT cells using tetramer and magnetic beads, we noted the same pattern, with all populations other than CD24^hi/CD44^lo/NK1.1^- cells being highly underrepresented (*Figure 1C,D*). These results are consistent with either a developmental block before Stage one or a severe defect in the survival or normal proliferation of iNKT cells from Stage one onwards. We also evaluated the prevalence of CD44/NK1.1-expressing T cells that were not tetramer-reactive, and noted a marked reduction in their numbers in liver and spleen as well (*Figure 1—figure supplement 2E–F*), suggesting a broad defect in the development of the NKT lineage.

To rule out cell-extrinsic mechanisms for this phenotype, we generated mixed hematopoietic chimeras reconstituted with a 1:1 mixture of wild-type (WT) and HDAC7-ΔP bone marrow. As we previously reported (*Kasler et al., 2012*), the HDAC7-ΔP transgenic population contributed robustly to the pool of CD4 and CD8 SP thymocytes, although there was a transient reduction in prevalence at the immature single positive (ISP) stage (*Figure 1—figure supplement 2A*). At early time points post-reconstitution (6-8wk), the distributions of naïve and memory T-cells in peripheral CD4 +and CD8+Tconv subsets were equivalent as well (*Figure 1—figure supplement 2B*). However, while the wild type-derived population reconstituted hepatic iNKT cells efficiently, HDAC7-ΔP bone marrow made almost no contribution to this compartment in the liver, where iNKT cells are most abundant (*Figure 1E*). This was also true in the thymus and spleen (*Figure 1—figure supplement 2C*), demonstrating that the abnormalities observed in the intact transgenic mice were due to a cell-autonomous mechanism.

We next examined the effects of loss of HDAC7 in the thymus on these phenotypes, using our previously characterized strain that deletes loxp-flanked *Hdac7* under the control of the *Lck* proximal promoter (*Hdac7^flox/:-::lck^cre*, henceforth *Hdac7-KO*) (*Kasler et al., 2011*). We previously reported that loss of HDAC7 during T cell development increased apoptosis of DP thymocytes leading to

inefficient positive selection. This shortened thymocyte lifespan resulted in a truncation of the TCR Jα repertoire, with distal rearrangements underrepresented (*Kasler et al., 2011*). It was thus not surprising to find that *Hdac7-KO* mice with an endogenous TCR repertoire had fewer iNKT cells than WT controls; for example, *Hdac7-KO* thymus had a 2–5 fold lower abundance of iNKT cells than WT littermates (*Figure 2A*). This reduction, consistent with the degree of underrepresentation of the relatively distal Jα18 TCR segment we previously noted (*Kasler et al., 2011*), was similarly observed in the spleen and liver (*Figure 3—figure supplement 1A,B*). Importantly, unlike the residual tetramer-reactive cells in HDAC7-ΔP mice, when staged after magnetic enrichment, iNKT calls in *Hdac7-KO* mice had normal expression of CD44 and NK1.1, suggesting that their development was not functionally altered. (Fig, 2A, at right).

Although deletion of Hdac7 did not result in expansion of NK1.1-expressing T cells, we did observe significant abnormalities in the effector programming of non-tetramer-reactive thymocytes. We noted a substantial expansion of a CD44hi Eomes +population in the mature CD8 SP compartment in the thymus (*Figure 2B*, *Figure 2—figure supplement 1B*). Examination of the peripheral CD8 T cells in these animals also showed a substantial increase in CD44 expression, suggesting an expansion of innate effector CD8 cells (*Figure 2C*). These cells resemble Eomes +innate memory CD8 +cells that are typically generated in trans, in response to IL4 secretion by thymic-resident iNKT cells (*Lee et al., 2013*; *Weinreich et al., 2010*), however as previously noted iNKTcells are depleted rather than expanded in *Hdac7-KO* mice, and we did not observe a consistent increase in the proportion of PLZF- or Vγ6.3-positive γδ T cells (*Figure 2—figure supplement 1A,C*), suggesting a different mechanism.

To clarify this question, we examined the phenotypes resulting from loss of HDAC7 in WT: *Hdac7-KO* mixed hematopoietic chimeras. In 1:1 chimeras, *Hdac7-KO* thymocytes competed equally through the ISP stage, but thereafter competed poorly and became steadily less abundant. This substantial underrepresentation of *Hdac7-KO* CD4 SP and mature CD8 SP thymocytes in 1:1 chimeras (*Figure 2—figure supplement 2A–B*) made analysis at this stage difficult, however their representation in the periphery was sufficient. In the spleen, we saw a strong increase in CD44 expression in the *Hdac7-KO*-derived vs. to the WT-derived CD8 T cell population (*Figure 2D,E*), duplicating what we saw in the intact mice and indicating that the phenotypes we observed are likely cell-autonomous. To further characterize the phenotype of these cells, we briefly stimulated splenocytes from these chimeras ex vivo, and found that *Hdac7-KO*-derived CD8 +T cells produced much more IFNγ than WT-derived CD8 +T cells in the same culture, assessed both as percent cytokine-positive (*Figure 3F,G*) and by median fluorescence intensity (MFI) of cytokine staining (*Figure 3H*). CD8 +T cells from *Hdac7-KO* population also had increased expression of the Eomes-associated chemokine receptor CXCR3 and the trafficking receptor Ly6C (*Figure 2—figure supplement 1D*).

Loss of HDAC7 thus appears to result in the aberrant adoption of innate effector programming by CD8 SP thymocytes that would otherwise have exited the thymus as naive Tconv. We observed a much more modest degree of abnormality in the CD4 compartment, comprising a 20–30% increase in the frequency of memory and IL4-secreting cells (*Figure 2—figure supplement 2B–C*), which we hypothesize is due to the greater similarity that CD8 thymic selection bears to NKT selection, in terms of both the similarity of CD1D to Class I MHC and the availability of selecting ligands on all thymocytes rather than just on specialized thymic APC. Loss of HDAC7 may thus allow some DP thymocytes to aberrantly adopt this lineage through some partial analogue of NKT selection. Collectively, our findings with both the *Hdac7-KO* and HDAC7-ΔP transgenic strains suggest that HDAC7 may function as a gatekeeper of innate effector programming, blocking the functional maturation of iNKT cells when constitutively expressed in the nucleus, and conversely allowing the aberrant acquisition of innate effector characteristics in Conventional T cells when it is conditionally deleted.

## HDAC7 regulates the effector programming of NKT cells in a manner that mirrors the function of PLZF

To generate a larger population of iNKT precursors for more in-depth evaluation the role of HDAC7, we employed the Vα14-Jα18 TCRα transgene (henceforth 'Vα14'), encoding the invariant TCRα chain that when paired with the appropriate endogenous β chains allows iNKT cells to bind glycolipids with high affinity (*Griewank et al., 2007*). Expressing this TCR transgene greatly increases the frequency of CD1D/αGalCer-reactive thymocytes, which arise naturally only at only around 1 in $10^4$ cells. As expected, mice expressing only the Vα14 transgene had many more iNKT cells in thymus

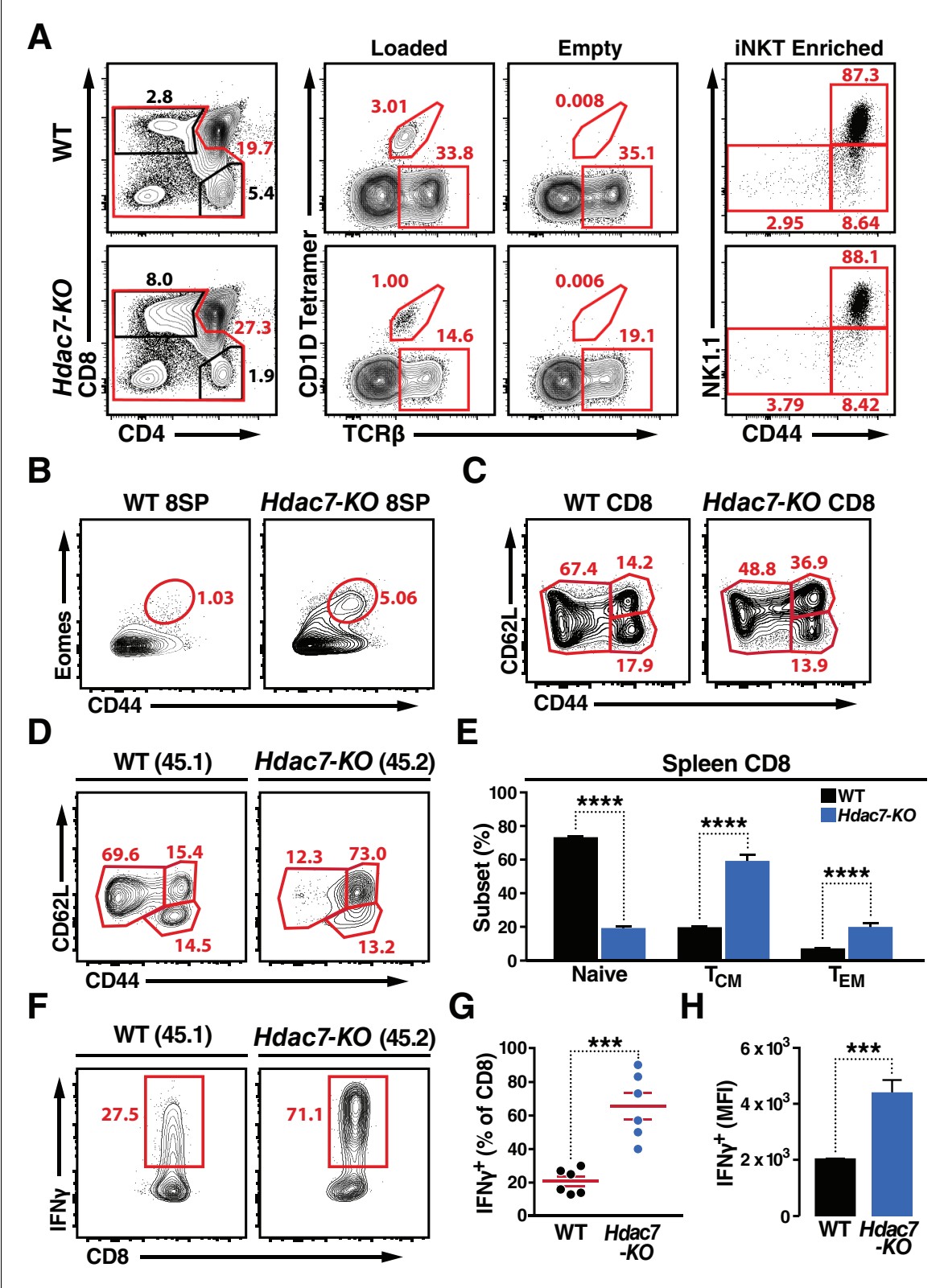

**Figure 2.** Deletion of HDAC7 in thymocytes Reduces iNKT Numbers and Expands an Innate-Memory CD8 Population. (**A**) Representative flow plots showing CD4/CD8 expression (left), loaded and empty CD1D tetramer reactivity (center), and CD44/NK1.1 expression of magnetically enriched iNKT cells (right) from thymus of WT (top) and *Hdac7-KO* (bottom) thymocytes. (**B**) Representative flow plots showing an expanded CD44$^{hi}$ Eomes$^+$ innate memory population in mature CD8SP thymocytes from *Hdac7-KO* mice. Mature CD8 SP thymocytes are identified as TCRβ$^+$CD8$^+$CD4$^-$. (**C**) Expression
*Figure 2 continued on next page*

*Figure 2 continued*

of CD44 and CD62L in CD8 T-cells from spleens of WT and *Hdac7-KO* littermate mice. Data are representative of 3 independent experiments with N = 2–4 mice per group. (D, E) Representative flow plots (D) and total quantification (E) of peripheral naive, central memory (T$_{CM}$), and effector memory (T$_{EM}$) CD8 T-cell populations from WT (CD45.1) and *Hdac7-KO* (CD45.2) derived bone marrow in mixed hematopoietic chimeras. (F, G, H) Representative flow plots (F) and total quantification (G, H) of IFNγ secretion in ex vivo stimulated CD8 T-cells. Splenocytes were harvested from mixed WT (CD45.1)/*Hdac7*-KO (CD45.2) hematopoietic chimeras, and stimulated ex vivo for 4 hr with PMA/Ionomycin. Percent of cells secreting IFNγ (G) and median fluorescence intensity (MFI) of IFNγ secretion (H) are shown. Bars on graphs indicate mean ±SEM (error bars). Data in (E) are combined from three independent experiments with at least three mice per group; data in (G, H) are combined from three independent experiments with two mice per group. Statistical significance was determined using either unpaired two-tailed T-test (E, H) or two-way ANOVA (G); ***p≤0.001, ****p≤0.0001. A Bonferroni post-test was used for pairwise comparisons in (E).
DOI: https://doi.org/10.7554/eLife.32109.008

The following source data and figure supplements are available for figure 2:

**Source data 1.** Microsoft Excel workbook containing numerical data matrices for all figure panels (on separate sheets) in which individual data points are not represented graphically.
DOI: https://doi.org/10.7554/eLife.32109.011
**Figure supplement 1.** Supporting Data on T Cell Phenotypes of *Hdac7-KO* Mice.
DOI: https://doi.org/10.7554/eLife.32109.009
**Figure supplement 2.** Supporting Data on Memory Markers and Cytokine Production in WT: *Hdac7-KO* Mixed Hematopoietic Chimeras.
DOI: https://doi.org/10.7554/eLife.32109.010

and spleen than WT mice (*Figure 3A*, *Figure 3—figure supplement 1A–B*). Also consistent with our expectations, when we crossed the Vα14 TCRα transgene into the *Hdac7-KO* strain, we observed a complete rescue of iNKT cell abundance in the thymus and periphery (*Figure 3—figure supplement 1A–B*), resulting in identical numbers between Vα14 and Vα14 *Hdac7-KO* mice. These cells were phenotypically similar to Vα14 iNKT cells in terms of CD44/NK1.1 expression (*Figure 3—figure supplement 1A*, bottom), suggesting that shortened thymocyte lifespan was indeed the main cause of the reduced iNKT abundance in *Hdac7-KO* mice.

In contrast to this finding, when the Vα14 transgene was co-expressed with HDAC7-ΔP, the rescue in the numbers of CD1D/αGalCer-reactive cells was incomplete (*Figure 3A*, *Figure 3—figure supplement 1C*), and the cells were phenotypically abnormal (*Figure 3B–F*). This result suggests that rather than blocking the maturation of CD1D/αGalCer-reactive cells categorically, HDAC7-ΔP blocked one or more steps normally associated with post-positive selection iNKT differentiation (*Benlagha et al., 2002*), directing the cells instead to mature as if they were positively selected Tconv. Consistent with this idea, other characteristics of CD1D/αGalCer-reactive Vα14 x HDAC7-ΔP T cells were similar to those of naïve Tconv. Flow analysis revealed that like the residual tetramer-reactive cells present in the HDAC7-ΔP mice (*Figure 1C*) the rescued iNKT cells in Vα14 x HDAC7-ΔP mice failed to upregulate the memory marker CD44 or the NKT marker NK1.1 in the thymus like their Vα14-only counterparts (*Figure 3C*, top row), although they did downregulate CD24 nearly as efficiently as Vα14 iNKT cells (*Figure 3—figure supplement 1D*), suggesting that they were able to mature to stage 1. This phenotype persisted in the spleen, after the HDAC7-ΔP transgene was turned off (*Figure 3B*, bottom row), suggesting that the cells had failed to complete effector programming in the thymus.

We next examined their cytokine responses to brief ex-vivo stimulation. When stimulated for 4 hr with PMA/ionomycin, CD1D/αGalCer-reactive WT and Vα14 transgenic iNKT cells exhibited a robust cytokine response, secreting both IFNγ and IL-4. In contrast, Vα14 x HDAC7-ΔP iNKT were far less likely to make IFNγ or IL-4 (*Figure 3C,D*), as would be expected for naïve Tconv. Additionally, iNKT cells typically express high levels of the integrin LFA-1 (CD11a/CD18), allowing them to remain localized in tissue-specific vascular beds such as hepatic sinusoids (*Thomas et al., 2011*). In contrast, Vα14 x HDAC7-ΔP iNKT cells exhibited far lower expression levels (*Figure 3E,F*), comparable to those seen in circulating non-CD1D/αGalCer-reactive CD4+ (mainly naïve) T-cells (*Figure 3F*, right). Moreover, while Vα14 x HDAC7-ΔP iNKT cells were found at comparable frequency in spleen to WT iNKT cells, they failed to concentrate in peripheral tissues such as the liver (*Figure 3—figure supplement 1E,F*), a behavior more characteristic of naïve Tconv rather than iNKT cells. These data are most consistent with a model in which HDAC7-ΔP prevents iNKT precursors from initiating innate effector development: Since Vα14 x HDAC7-ΔP iNKT cells have low CD44 expression, produce few

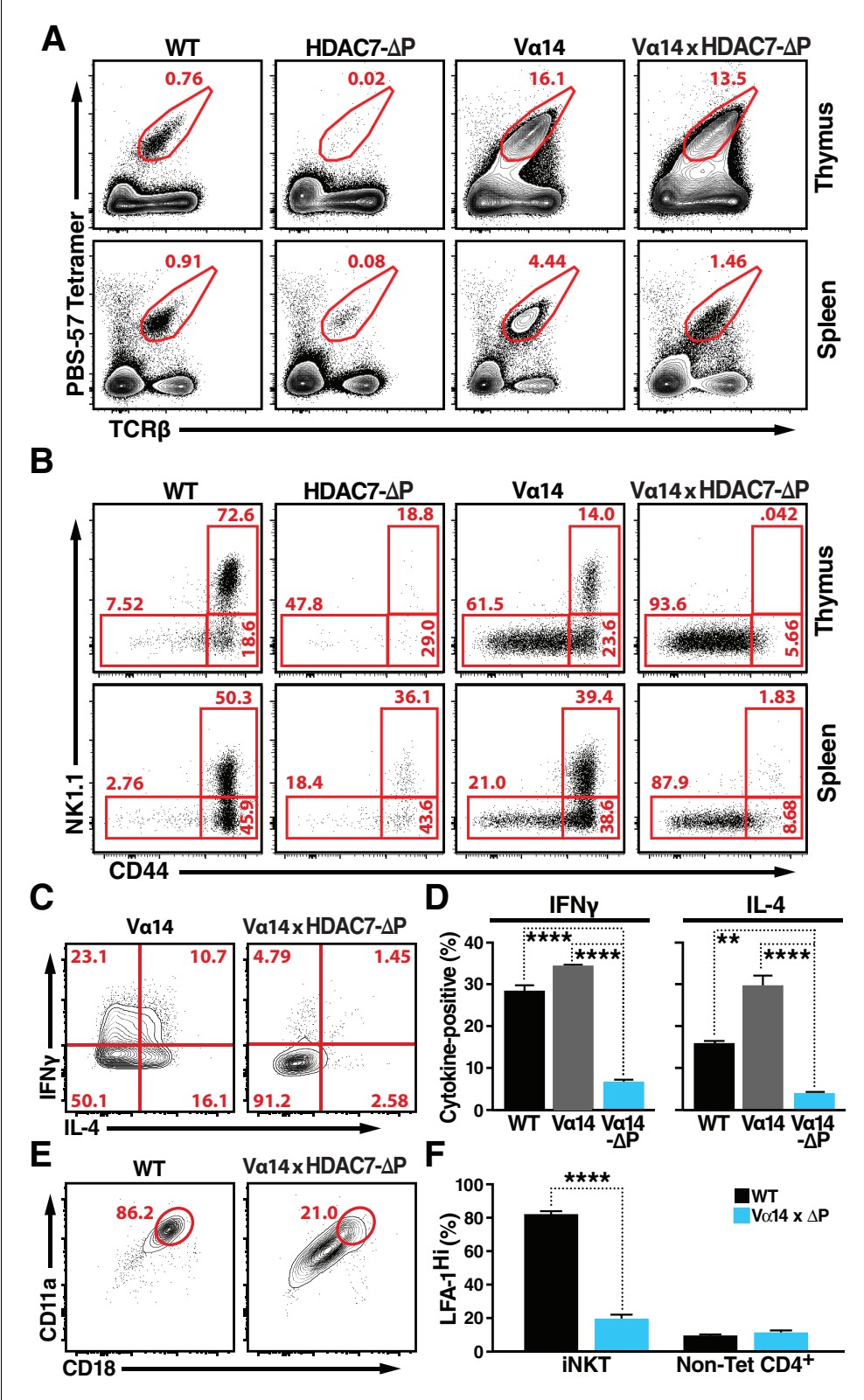

**Figure 3.** HDAC7-ΔP Blocks Innate Effector Development in iNKT Cells and Converts Them to Naive-Like T-cells. (A, B) Representative flow cytometric plots showing TCRβ vs. PBS-57 tetramer staining (A), and CD44 vs. NK1.1 staining of iNKT cells (B) in thymus (top) and spleen (bottom) of littermate mice with the indicated genotypes. (C, D) Representative staining (C) and total quantification (D) of IFNγ and IL-4 secretion in total splenocytes from littermate mice of the indicated genotypes, stimulated ex vivo for 4 hr with PMA/Ionomycin. (E, F) Representative flow plots (E) showing surface

*Figure 3 continued on next page*

Figure 3 continued

expression of LFA-1 (CD11a/CD18) in splenic iNKT (Tet$^+$/TCRβ$^+$) cells, with quantification for four littermate pairs shown in (**F**). Bars on graphs indicate mean ±SEM. Data in (**D, F**) are from three independent experiments. Statistical significance was determined using one-way (**E**) or two-way (**F**) ANOVA; *p≤0.05, **p≤0.01, ***p≤0.001, ****p≤0.0001. Tukey (**E**) or Bonferroni post-tests (**F**) were used for pairwise comparisons.

DOI: https://doi.org/10.7554/eLife.32109.012

The following source data and figure supplement are available for figure 3:

**Source data 1.** Microsoft Excel workbook containing numerical data matrices for all figure panels (on separate sheets) in which individual data points are not represented graphically.

DOI: https://doi.org/10.7554/eLife.32109.014

**Figure supplement 1.** Supporting Data on iNKT Phenotype of Vα14/Jα18 X HDAC7-ΔP Transgenic Mice.

DOI: https://doi.org/10.7554/eLife.32109.013

cytokines after brief stimulation, and freely recirculate, they appear to become diverted into functionally naïve-like T-cells.

When considering how both gain and loss of thymic HDAC7 function alter innate effector development, we were struck by how closely our results mirrored findings reported in similar studies of the transcription factor PLZF. Specifically, the severe depletion of iNKT cells (*Figure 1A*) and loss of effector memory phenotype in peripheral iNKT cells observed in gain-of-function HDAC7-ΔP (*Figure 3B*) strongly resembles the iNKT defect observed in PLZF knockouts (*Kovalovsky et al., 2008*; *Savage et al., 2008*). Conversely, the consequences of loss of HDAC7 function – notably expansion of IFNγ-secreting CD8 +and IL4-secreting CD4 +memory cells (*Figure 2D–H*, *Figure 1— figure supplement 2B–C*) mirror results reported in gain-of-function PLZF transgenic mice (*Kovalovsky et al., 2010*; *Savage et al., 2011*). Polyclonal (non-invariant) type II NKT cells are also thought to be PLZF-dependent (*Zhao et al., 2014*), and we similarly noted a near absence of tissue-resident type II NKTs in HDAC7-ΔP mice, defined by a Tet$^-$TCRβ$^+$CD8$^-$CD44$^{hi}$NK1.1$^+$ profile (*Figure 1—figure supplement 1E–F*). HDAC7 and PLZF thus appear to play nearly inverse roles in iNKT development (*Figure 4E*).

One possible mechanism for this inverse relationship is that nuclear HDAC7 represses the expression of PLZF, preventing HDAC7-ΔP thymocytes from expressing PLZF (*Seiler et al., 2012*). Indeed, we observed a pronounced reduction in PLZF expression in TCRβ+T cells from HDAC7-ΔP mice in all organs examined, including thymus, spleen and liver (*Figure 4A,B*). However, PLZF was still detected in CD4 +SP thymocytes from Vα14 x HDAC7-ΔP mice; although expression was restricted compared to Vα14-only thymus (*Figure 4C*), Interestingly, PLZF expression was maintained in roughly half of splenic Vα14 x HDAC7-ΔP iNKT cells (*Figure 4D*, right panel). Thus, transcriptional repression of PLZF expression by HDAC7 is probably insufficient to fully explain the iNKT phenotype, as even PLZF +Vα14 x HDAC7-ΔP iNKT cells exhibit naïve-like characteristics (*Figure 3B*).

## HDAC7 and PLZF inversely regulate a shared innate effector gene network that is highly relevant to autoimmune disease

The remarkable inverse relationship between the phenotypes mediated by alterations of HDAC7 and PLZF function in iNKT cell development prompted us to take an unbiased, genome-wide approach to understanding how these two factors might coordinately regulate the transcriptional landscape of this process. To this end, we generated gene expression profiles by RNA-seq of PBS-57 tetramer-reactive Vα14 Tg and Vα14 X HDAC7-ΔP Tg thymocytes and splenocytes, as well as polyclonal naïve (i.e. CD44$^-$) conventional CD4 SP thymocytes and splenocytes. Differential gene expression profiles were constructed for Vα14 Tg vs. naïve Tconv, Vα14 X HDAC7-ΔP Tg vs naïve Tconv, and Vα14 X HDAC7-ΔP vs. Vα14 Tg, by comparing the normalized scalar expression values for three biological replicates of each condition, based on roughly 40 million mapped reads per sample (See *Supplementary file 1*, Materials and methods). When we plotted significant expression changes for tetramer-reactive Vα14 Tg cell vs. Tconv (*Figure 5A*, left and right panels, horizontal axes) against the corresponding changes for Vα14 X HDAC7-ΔP vs Tconv (vertical axes), it was evident in both thymus and spleen that HDAC7-ΔP makes the overall gene expression pattern of tetramer-reactive cells more similar to that of Tconv, as shown by the clockwise shift of the plot trend line from the diagonal in both tissues (*Figure 5A*, solid plot diagonal vs. dotted trend line). Reflecting this effect, iNKT development-associated gene expression changes (both up and down) that were

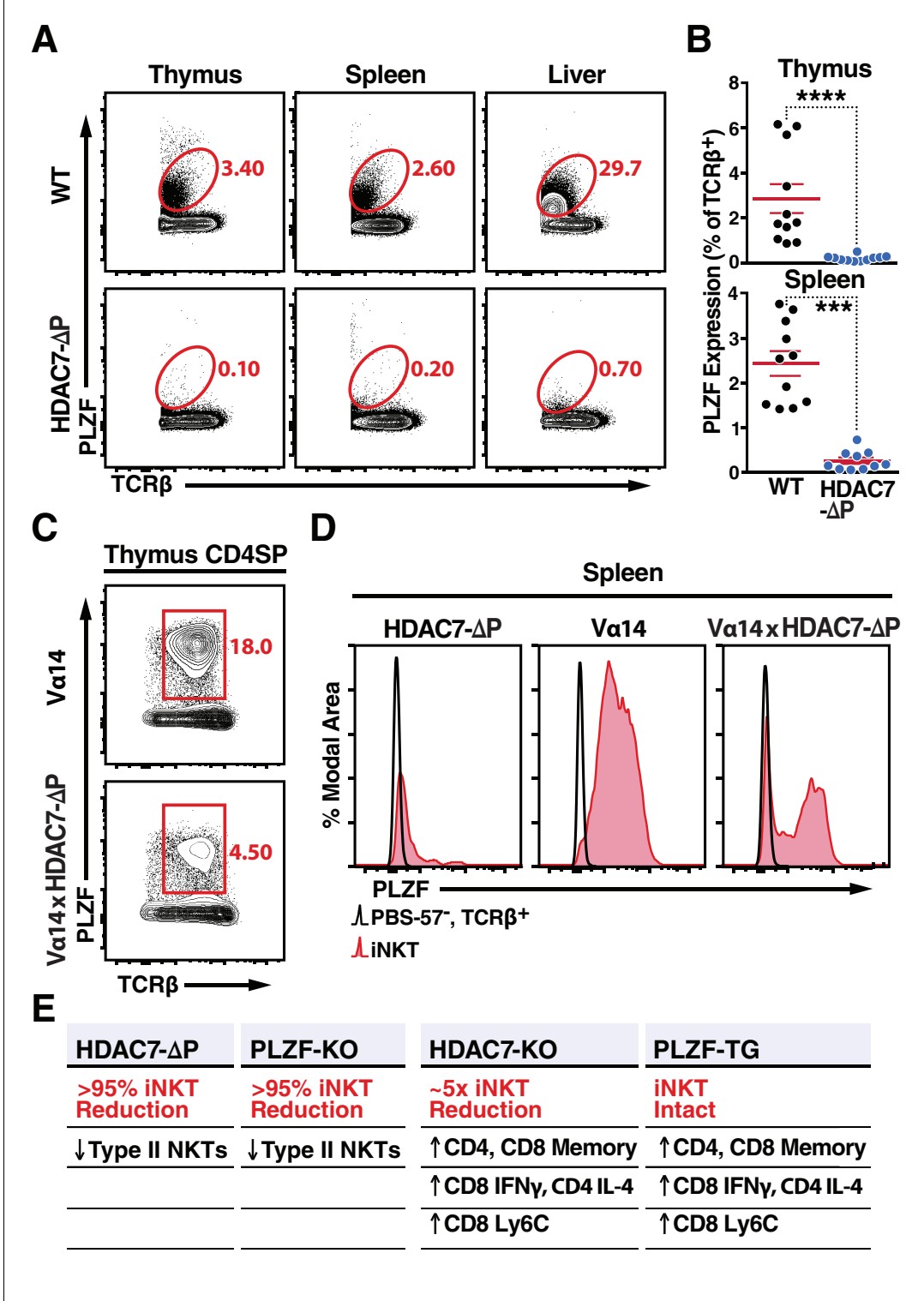

**Figure 4.** Nuclear HDAC7 Retention Restricts PLZF Expression and Mirrors PLZF-Associated T Cell Phenotypes. (**A, B**) Representative flow cytometric plots (**A**) and total quantification (**B**) of PLZF expression in TCRβ+ cells from thymus, spleen, and liver. (**C**) PLZF expression in mature CD4 SP (CD4+ CD8- TCRβ+) thymocytes from Vα14 (top) and Vα14 X HDAC7-ΔP (bottom) transgenic animals. (**D**) PLZF expression in peripheral iNKT (Tet+ TCRβ+) cells from spleen. Black unfilled histograms correspond to conventional (Tet-TCRβ+) T-cells, red tinted to iNKT (Tet+TCRβ+) cells. (**E**) Summary table comparing phenotypes in HDAC7-ΔP, PLZF KO, *Hdac7-KO* and PLZF Tg mice with respect to iNKT and conventional T-cell development. Bars on graphs in (**B**) indicate mean ±SEM; symbols represent individual mice. Data in (**B**) are combined from four independent experiments with at least two
*Figure 4 continued on next page*

*Figure 4 continued*

mice per group; data in (D) are representative of 3 independent experiments with two mice per group; Statistical significance was determined using unpaired two-tailed T-tests (B); ***p≤0.001, ****p≤0.0001 vs. WT.

DOI: https://doi.org/10.7554/eLife.32109.015

suppressed by HDAC7-ΔP (*Figure 5A*, green plot points and numbers) greatly outnumbered those enhanced by HDAC7-ΔP (*Figure 5A*, red plot points and numbers) in both spleen and thymus. Strongly induced genes involved in iNKT cell development that were suppressed by HDAC7-ΔP included *Id2*, *Zbtb16* (PLZF), *Klrb1c* (NK1.1), *Tbx21* (T-bet), *Gata3*, *Il4*, *Ifng*, and *Zfp683*, which encodes HOBIT, a zinc-finger transcription factor recently shown to be essential for the acquisition of tissue-resident effector function (*Mackay et al., 2016*) (*Figure 5A*, labeled points). This pattern of suppression was established in the thymus (*Figure 5A*, left), but persisted in the spleen (*Figure 5A*, right), after expression of HDAC7-ΔP was turned off. Blocking HDAC7 nuclear export in the thymus thus apparently programs a more naïve-like state of differentiation into tetramer-reactive cells that persists even after HDAC7-mediated repression is removed. Although some of the changes in gene expression that we observe, especially in the spleen, may be due to the different population distributions with respect to conventional iNKT staging that were sampled between the Vα14 Tg and Vα14 X HDAC7-ΔP tetramer-reactive cells, for many of the genes we identified (e.g. Hobit, T-bet, *Figure 5A*), the magnitude of the suppression, i.e. lower than in the WT cells, is still greater than could be accounted for by this explanation.

These data were also helpful in identifying key candidate molecular targets of HDAC7. Ingenuity Pathway Analysis (IPA, Qiagen) analysis of putative upstream regulators of the HDAC7-affected gene set identified multiple targets highly relevant to iNKT development and function, including *Zbtb16* (PLZF), *Id2*, *Il4*, *Ifng*, *Tbx21* (T-bet), and *Gata3* (*Figure 5—figure supplement 1A*, for a complete list of putative upstream regulators see *Supplementary file 3*). The downstream targets of these were almost universally affected in a manner that suggests inhibition rather than activation of the putative upstream regulator (*Figure 5—figure supplement 1a*, column 2). The expression of most of these upstream regulators was itself suppressed by HDAC7, suggesting an obvious mechanism of regulation (*Figure 5—figure supplement 1A*, column 3), however the Tec kinase *Itk*, the most highly correlated upstream regulator of HDAC7 targets in both thymus and spleen, was only modestly suppressed in spleen and not significantly suppressed in thymus, suggesting that HDAC7 might regulate its activation more than its expression. ITK has a well-characterized role in the maturation of conventional CD8 T cells, CD8 innate effectors, and iNKT cells (*Atherly et al., 2006*; *Felices and Berg, 2008*).

Similarly, PLZF (*Zbtb16*) expression was relatively modestly repressed by HDAC7-ΔP (e.g, 12-fold, vs. 30 fold induction in in thymus, *Figure 5A*), yet its downstream targets were very highly correlated with the HDAC7 target gene set, based on both IPA analysis and comparison of HDAC7-regulated genes with genes identified in a recent, comprehensive study of PLZF-regulated genes in iNKT cell development (*Mao et al., 2016*) (*Figure 5B–C*, *Figure 5—figure supplement 1C*). Gene expression changes due to loss of PLZF function in iNKT cells show a clear positive correlation with changes caused by expression of HDAC7-ΔP (*Figure 5B*, *Figure 5—figure supplement 1C*), while changes caused by expression of a PLZF transgene show a clear negative correlation (*Figure 5B*, *Figure 5— figure supplement 1C*), demonstrating an inverse relationship between HDAC7 and PLZF function. Genes that were found to associate directly with PLZF by chIP-seq (*Mao et al., 2016*), cluster strongly around the HDAC7-PLZF diagonals, and are highly concentrated among the most negatively correlated genes in terms of the effects of HDAC7 vs. PLZF function (*Figure 5D–E*, labeled genes, *Figure 5—figure supplement 1C*, red asterisks). Out of the 31 genes reported by Mao, et al. to be both bound by PLZF and differentially expressed in iNKT cells due to alteration of PLZF function, 17 were found on the PLZF-HDAC7 inverse diagonals and only four on the positive diagonals (*Figure 5B–C*, labeled genes, *Figure 5—figure supplement 1C*, red asterisks). An additional four genes were negatively correlated with HDAC7 function but not differentially expressed during iNKT development (see *Supplementary file 1*), while one was positively correlated. Additionally, Mao, et al. identified BACH2 as a crucial interaction partner of PLZF, and our own data show BACH2 as not differentially expressed but nonetheless as one of the strongest putative upstream regulators of

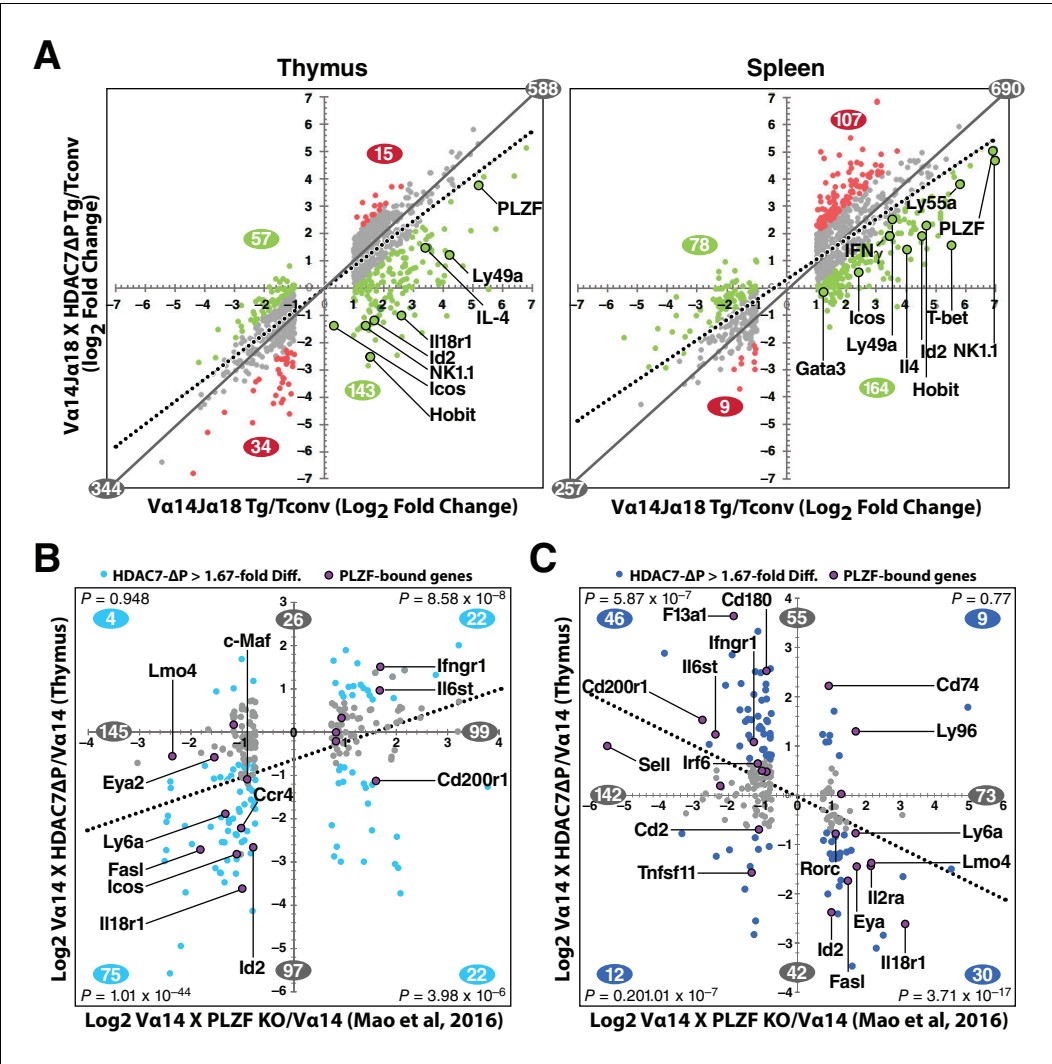

**Figure 5.** HDAC7 Regulates a Cassette of Genes in Glycolipid-Reactive Cells That is Highly Relevant to Innate Effector Function, Inflammation, Autoimmunity, and Autoimmune Liver Disease. (**A**) Scatter charts showing gene expression changes in Cd1d/αGalCer-reactive Vα14 Tg (X axis) and Vα14 X HDAC7-ΔP Tg (Y axis) thymocytes (left) or CD4 splenocytes (right) vs naïve CD4SP thymocytes or splenocytes, respectively. The solid gray line indicates the plot diagonal and the dotted gray line indicates the Least Squares best-fit line of the plotted data. Genes displayed were expressed at least 1.75-fold differentially between tetramer-reactive and naïve cells, with p<0.05 (2-tailed Student's T test) for three biological replicates of each genotype. Colored plot points represent genes whose differential expression vs. naïve was enhanced (red points) or suppressed (green points) at least 1.75-fold by co-expression of HDAC7-ΔP (**C, D**) Scatter charts showing genes > 1.66 fold differentially expressed due to loss of PLZF function (C, horizontal axis), or due to transgenic expression of PLZF (D, horizontal axis) according to (*Mao et al., 2016*), plotted against effect of HDAC7-ΔP expression in PBS-57 tetramer-reactive Vα14 X HDAC7-ΔP vs Vα14 Transgenic thymocytes in Thymus (C, vertical axis) or spleen (D, vertical axis). Total number of genes > 1.67 fold differentially expressed along each axis are indicated in gray. Numbers of genes, with P-values (binomial distribution) of the overlap, for genes differentially expressed along both axes in each quadrant (blue symbols), are indicated in blue.

DOI: https://doi.org/10.7554/eLife.32109.016

The following figure supplement is available for figure 5:

**Figure supplement 1.** HDAC7 Regulates a Cassette of Genes in Glycolipid-Reactive Cells That is Highly Relevant to Innate Effector Function, Inflammation, Autoimmunity, and Autoimmune Liver Disease.

DOI: https://doi.org/10.7554/eLife.32109.017

the HDAC7-regulated gene set (*Figure 5—figure supplement 1A*), suggesting that HDAC7 may modulate its targets via a ternary interaction with PLZF. This remarkable degree of overlap strongly supports the idea that HDAC7 is a negative regulator of iNKT cell development that functions at least in part by negatively regulating PLZF-dependent transcription.

Ontologic analysis of HDAC7-regulated genes using IPA provided strong evidence for their association with both innate-like effector function and inflammatory disease. Canonical pathways associated with the HDAC7-regulated gene set included multiple pathways associated with innate immune signaling and T cell effector function (*Figure 5—figure supplement 1B*, green-shaded pathways, see *Supplementary file 2* for a complete list of pathways and associated genes), as well as with inflammation and inflammatory disease states (*Figure 5—figure supplement 1A*, blue-shaded pathways), particularly hepatic inflammation. This connection was brought into even sharper relief by two recent GWAS studies of primary sclerosing cholangitis (PSC) and inflammatory bowel disease (IBD), which both identified HDAC7 among the disease-associated loci, and also individually its immediate upstream kinases PKD and SIK2, as well as two isoforms of PKC that are upstream of PKD (*Figure 6A*) (*Liu et al., 2013*; *Jostins et al., 2012*). Moreover, a remarkably high proportion of the other hits from these studies are downstream of HDAC7, i.e. their expression in iNKT cells is altered by HDAC7-ΔP. Of the 176 GWAS risk loci mapping to genes that were expressed in our RNA-seq data, 81 (46%) were regulated by HDAC7 in NKT cells, a much higher degree of overlap than would be expected by chance (p=$3.49 \times 10^{-16}$, binomial distribution) (*Figure 6A*). Of the 16 strongest risk loci identified by the Liu, et al. study of PSC, 10 were differentially expressed due to expression of HDAC7-ΔP, and four more comprised HDAC7 itself, as well as its upstream regulators PRKD2 and SIK2, and also PLZF interaction partner BACH2 (*Parra et al., 2005*; *Mao et al., 2016*; *Liu et al., 2013*) (*Figure 6*).

To gain a better understanding of the significance of this overlap, we evaluated the 81 risk loci that were regulated by HDAC7 with respect to regulation by PLZF, differential expression in iNKT cells vs. Tconv, functional role in iNKT cell development, and functional role in autoimmune disease (*Figure 6B*). This analysis revealed that a large proportion of these genes were functionally important in iNKT development (*Figure 6B*, sixth row), while relatively fewer were identified as PLZF targets (*Figure 6B*, third row), suggesting that HDAC7 affects autoimmunity and iNKT development via both PLZF-dependent and independent mechanisms. We then further filtered the genes for significance in at least 4 of the eight criteria examined (*Figure 6B*), and then manually mapped the resulting 56 genes to their associated signaling pathways (*Figure 6C*). Remarkably, all but 13 of these genes could be mapped to one of five interconnected signaling networks, comprising Th1 and Th2 cytokine signaling, chemokine signaling, TCR signaling with its associated costimulatory pathways, and signaling through cell membrane-associated TNF superfamily members (*Figure 6C*, dark-colored symbols with white label, gray-shaded areas).

These signaling networks are also heavily populated with HDAC7 targets that were not identified in the GWAS studies (*Figure 6C*, light-colored symbols), an observation that is confirmed by IPA analysis of canonical signaling pathways and upstream regulators among HDAC7 targets (*Figure 5—figure supplement 1A,B*). In nearly all cases, HDAC7 regulates these targets in a manner opposite to their regulation during iNKT cell development (*Figure 6C*, symbol border vs. fill colors). This regulation by HDAC7 by clearly suppresses downstream signaling in all cases except for TNF superfamily costimulatory signaling, which is mostly potentiated (*Figure 6C*, color of arrows per legend). Consistent with our phenotypic findings, nearly half of these genes have positive roles in NK/NKT development/function (*Figure 6B*), showing that HDAC7 broadly suppresses several key signaling pathways that are highly important in both NKT cells and in human autoimmune diseases that are similar to the pathology observed in HDAC7-ΔP transgenic mice. This remarkable concordance strongly supports the idea that the role of HDAC7 in these cells is involved in the pathogenesis of PSC and IBD, and identifies a few key signaling pathways as candidates for further interrogation.

## HDAC7 physically binds to PLZF and modulates its transcriptional activity

HDAC7 is a class IIA histone deacetylase that lacks intrinsic DNA binding capacity and requires binding to target transcription factors to modulate transcription at specific loci (*Yang and Seto, 2008*). Class IIA HDACs typically act as dominant corepressors, as in the case of MEF2, which is converted from a transcriptional activator to a repressor upon class IIA HDAC binding (*McKinsey et al., 2000*).

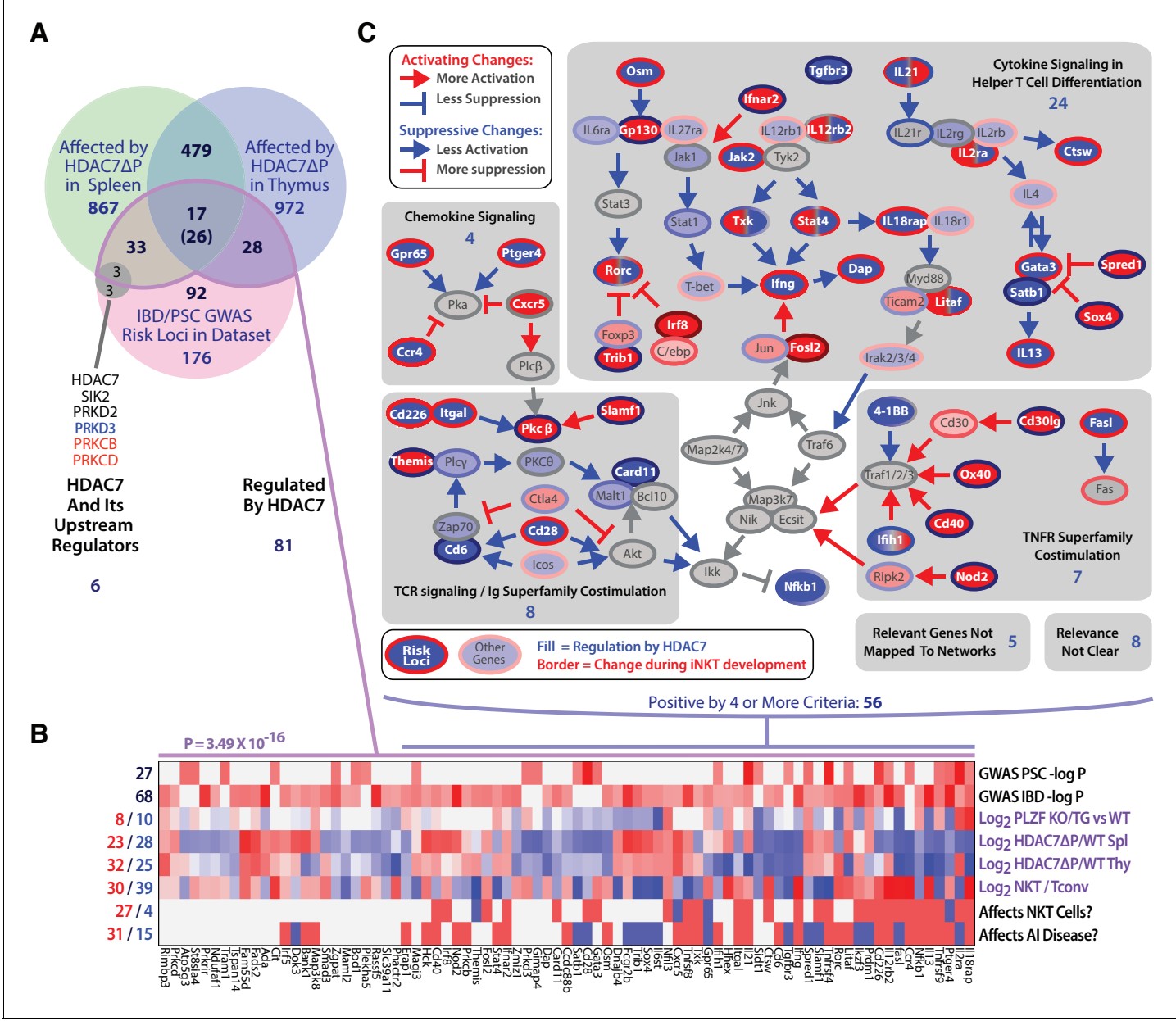

**Figure 6.** The intersection of HDAC7-regulated genes in iNKT development and GWAS hits for IBD and PSC highlights key signaling pathways. (**A**) Venn diagram showing enumeration of genes that are GWAS risk loci for PSC and IBD from (*Liu et al., 2013*; *Jostins et al., 2012*), and/or also regulated by HDAC7 during NKT development according to our RNA-seq data (FDE >1.66, p<0.05). The indicated P-value is based on the binomial distribution, using the 13,519 genes scored as expressed under any condition as a basis. (**B**) Heatmap showing the P-values for the cited GWAS studies in the overlapping set of genes (first two rows), regulation of these genes by PLZF (third row, according to [*Mao et al., 2016*]), by HDAC7 (rows 4–5), and during normal iNKT development, (row 6, according to Immgen stage-specific data [http://www.immgen.org]), as well as scoring for positive (red) or negative (blue) roles in NK/NKT development/function or autoimmunity, according to literature search (rows 7–8, see *Supplementary file 1* for citations). (**C**) Genes that were most relevant to the criteria listed in (**B**), i.e. positive/significant in four or more measures, were mapped the signaling pathways in which they participate. Shaded areas indicate four distinct, highly populated signaling modules, with number of genes indicated.
DOI: https://doi.org/10.7554/eLife.32109.018

PLZF belongs to the BTB-ZF family of transcription factors (*Beaulieu and Sant'Angelo, 2011*) previously reported to interact with class IIA HDACs (*Verdin et al., 2003*; *Chaucherau et al., 2004*); indeed, one group has even demonstrated in vitro and in vivo binding of HDAC7 to PLZF in a separate cell type (*Lemercier et al., 2002*). This suggested that HDAC7 might modulate PLZF activity in

thymocytes through direct physical binding. Determining if this is the case directly is somewhat challenging however, as the abundance of PLZF in wild-type thymocytes is very low, being restricted to a small population of iNKT precursors. To circumvent this difficulty, we made cell lysates from PLZF-transgenic thymocytes and immunoprecipitated them with antibodies to endogenous HDAC7. These experiments showed a specific interaction between HDAC7 and PLZF in thymocytes (*Figure 7A*).

To further define this interaction, we co-transfected FLAG-tagged full-length or truncated HDAC7 with full length HA-tagged PLZF (*Figure 7B,D*), or conversely different truncations of PLZF with the (interacting) HDAC7 N-terminal adapter domain (residues 1–497, *Figure 7C,E*). After Immunoprecipitation of transfected lysates with anti-FLAG agarose beads, we quantified the amount of PLZF protein pulled down vs. input levels over 3–6 separate experiments for each construct, using the LiCor Odyssey system (*Figure 7D–E*). The results of this analysis identify residues 65–200 of HDAC7, containing the MEF2-interacting domain through the first PKD phosphorylation site, as the interacting region (*Figure 7D*). Analysis of the PLZF deletions identified a region from residues 320–450, encompassing a proline-rich tract and the first two zinc finger domains, as critical for interaction (*Figure 7E*).

Although the precise mode of transcriptional regulation by PLZF remains unclear, with different domains exhibiting activating and repressive activity in varying contexts (*Sadler et al., 2015*; *Puszyk et al., 2013*; *Melnick et al., 2002*), we next wanted to examine if HDAC7 physical binding to PLZF could modulate its transcriptional activity. We transfected 293 T cells with fusions of the GAL4 DNA binding domain (residues 1–142) to full-length PLZF, an HDAC7-interacting mutant of PLZF (1-460), or a non-interacting mutant (1-318), together with a SV40 minimal promoter-Gal4(5)-firefly luciferase reporter and an EF-1α promoter-driven *Renilla* luciferase reporter (*Figure 7F*, see *Figure 7—figure supplement 71* for a diagram). To these constructs were added empty vector (*Figure 6F*, light blue bars), a vector encoding full-length HDAC7 (*Figure 7F*, dark blue bars), or one encoding a fusion of the HDAC7 1–497 interacting domain with the VP16 transcriptional activation domain (HDAC7-VP16, *Figure 7F*, medium blue bars). Measurement of luciferase activity in lysates from these cells showed that co-transfection of FL HDAC7 with FL PLZF or the 1–460 truncation reduced transcription from the Gal4-luc construct, while it did not affect transcription when co-transfected with the non-interacting 1–318 mutant (*Figure 7F*). Conversely, HDAC7-VP16 increased transcription from the interacting PLZF constructs but not the non-interacting one (*Figure 7F*). These experiments, together with our characterization of the HDAC7-PLZF interaction and transcriptional targets above, provide strong evidence that in thymocytes HDAC7 regulates PLZF in the same manner as MEF2 and other transcription factors, functioning as a TCR signal-dependent co-repressor that helps to silence PLZF-associated promoters in the absence of appropriate signals. This mechanism is likely to account for at least part of the effect of HDAC7 on iNKT cells.

## Restoring iNKT cells ameliorates Tissue-Specific autoimmunity

We earlier reported that HDAC7-ΔP mice develop spontaneous tissue-specific autoimmunity, with about 80% developing obliterative exocrine pancreatitis and concomitant T-cell infiltration in stomach, liver and small intestine within eight months (*Kasler et al., 2012*). Although this had been previously attributed solely to a defect in negative selection of conventional thymocytes, the striking absence of iNKT cells in HDAC7-ΔP mice spurred us to consider whether disrupted innate effector development might also contribute to this autoimmune syndrome. Indeed, the very tissues vulnerable to T-cell infiltration in HDAC7-ΔP mice, notably the small intestine, liver and hepatobiliary mucosa, are typically populated by PLZF-dependent innate effectors such as iNKT and mucosal-associated invariant T (MAIT) cells (*Fan and Rudensky, 2016*). We thus set out to determine if restoring iNKT cells could alter the course of HDAC7-ΔP–induced autoimmunity.

In our earlier studies, we found that that HDAC7-ΔP-mediated autoimmunity is dominantly transferable in mixed BM chimeras if a 5-fold excess of HDAC7-ΔP-derived bone marrow is used. While engraftment at these ratios produced comparable populations of WT and HDAC7-ΔP Tconv in peripheral tissues, we did not assess the reconstitution of the iNKT compartment in those studies (*Kasler et al., 2012*), leaving open the possibility that there was an uncharacterized recessive component to the autoimmunity. Attempts to adoptively transfer mature iNKT cells directly into HDAC7-ΔP mice failed to effectively restore tissue-resident iNKT populations (*Figure 8—figure supplement 81A–C*). Instead, we generated two sets of hematopoietic chimeras to determine if restoring iNKT cells using Vα14 bone marrow could ameliorate disease compared to WT bone marrow (*Figure 8A*).

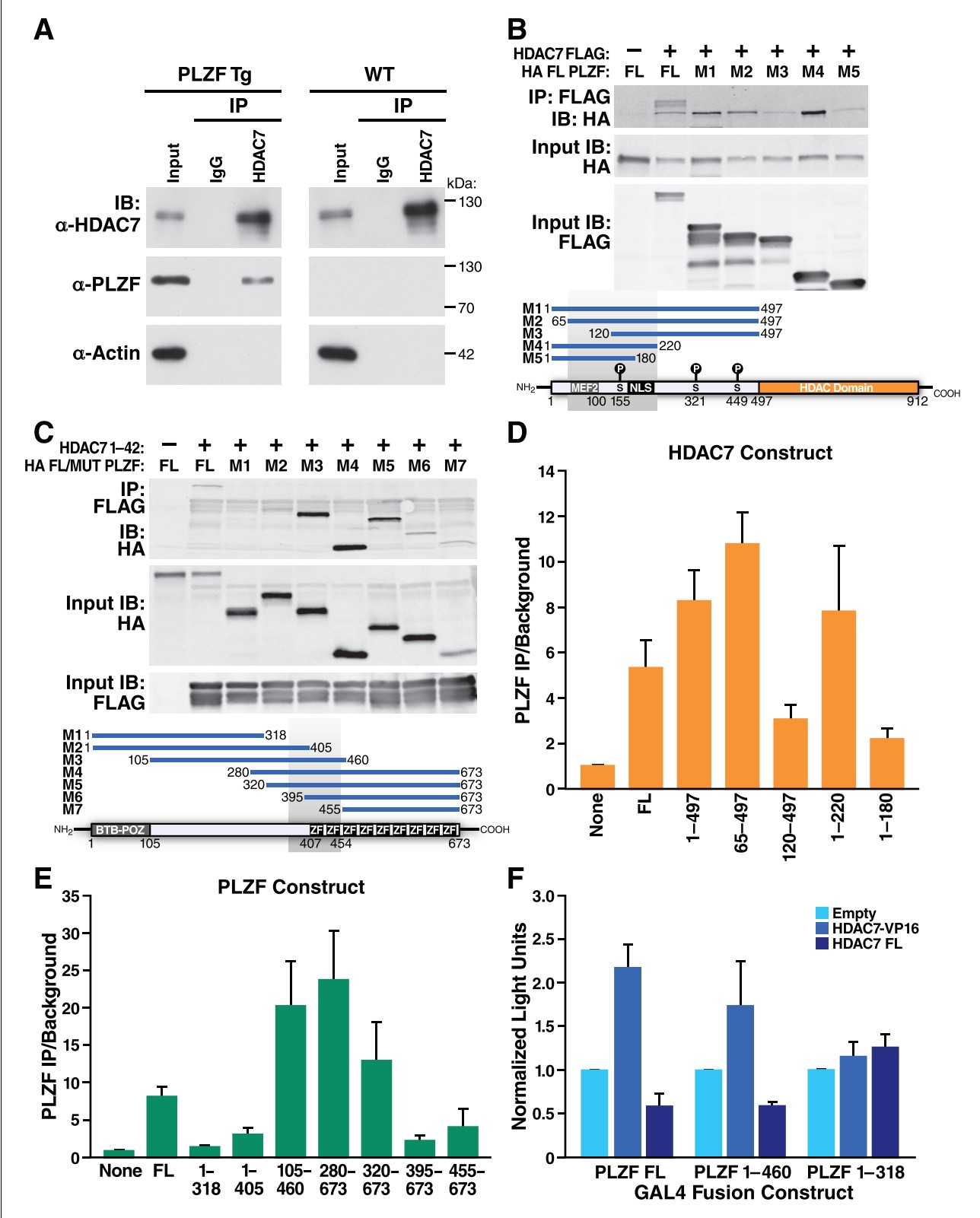

**Figure 7.** HDAC7 Can Physically Bind and Functionally Antagonize PLZF Transcriptional Activity. (**A**) Immuno-blots showing co-immunoprecipitation with endogenous HDAC7 of PLZF from PLZF-transgenic thymocytes. (**B**) Immunoblot showing Co-immunoprecipitation of HA-tagged full-length PLZF from transfected 293 T cells with the indicated FLAG-tagged truncation mutants of HDAC7 (**C**) Immunoblot showing Co-immunoprecipitation of HA-tagged PLZF truncations as indicated, with the FLAG-tagged HDAC 1–497 (**D**), (**E**) Quantification of Immunoprecipitated protein/input protein for the

*Figure 7 continued on next page*

*Figure 7 continued*

pairs of constructs in (B) and (C) respectively. Ratios shown are normalized to the background signals for each individual experiment. Error bars indicate SEM of 4–7 individual experiments for each pair of constructs. Shaded areas in diagrams in (B) and (C) indicate areas defined as required for interaction based on this analysis. (F) Firefly luciferase activity from 293 T cells transfected with a Gal4(5)/SV40 minimal promoter reporter construct, normalized to *Renilla* luciferase values from an EF1α promoter-driven reporter construct. In addition to the reporters, cells were transfected with constructs encoding the Gal4 DNA-binding domain (1-142) fused to the indicated segments of PLZF, as well as empty vector, full-length HDAC7, or HDAC7 1–497 fused to the HSV VP16 transcriptional activation domain (410-490). Error bars represent SEM of four individual experiments.
DOI: https://doi.org/10.7554/eLife.32109.019

The following source data and figure supplement are available for figure 7:

**Source data 1.** Microsoft Excel workbook containing numerical data matrices for all figure panels (on separate sheets) in which individual data points are not represented graphically (*Figure 7D,E,F*).
DOI: https://doi.org/10.7554/eLife.32109.021

**Figure supplement 1.** Diagram of GAL4-PLZF, HDAC7, and GAL4 reporter constructs employed in experiments shown *Figure 7F* in main text.
DOI: https://doi.org/10.7554/eLife.32109.020

When irradiated recipients were reconstituted with a 1:5 mixture of Vα14: HDAC7-ΔP bone marrow, peripheral iNKT cells were effectively rescued to normal levels, while in recipients receiving a 1:5 WT: HDAC7-ΔP mixture they were still essentially absent (*Figure 8B*).

Comparing these cohorts over time, we noted Vα14: HDAC7-ΔP chimeras had significantly lower peak plasma levels of ALT and AST, commonly used as an indication of liver damage, than WT: HDAC7-ΔP chimeras (*Figure 8C*). Both cohorts eventually perished from exocrine pancreatitis and had similar pancreatic lipase levels in plasma (*Figure 8—figure supplement 81D*), yet Vα14: HDAC7-ΔP chimeras exhibited significantly improved body weight maintenance in the first two months post-engraftment (*Figure 8D*, left) and a reduced overall mortality rate (*Figure 8D*, right) compared to WT: HDAC7-ΔP chimeras. These results provide evidence that disruptions in innate effector development, particularly the loss of iNKT cells in the hepatobiliary tract, exacerbates tissue specific autoimmunity in the HDAC7-ΔP setting. Restoring this missing innate effector population resulted in enhanced survival and a significant reduction in the severity of disease.

## Discussion

### HDAC7 nuclear export licenses innate effector development

The discovery and characterization of innate effector lymphocytes has transformed our understanding of T-cell receptor signaling, barrier protection at mucosal surfaces, and the evolutionary origins of the vertebrate immune system, yet the identification of key regulatory factors that control naïve versus innate effector development in thymocytes is far from complete. We demonstrate here that the epigenetic regulator HDAC7 serves as a gatekeeper of this developmental fate decision in the thymus. When HDAC7 is prevented from releasing its genomic targets in response to TCR stimulation, PLZF-dependent innate effector development appears to be blocked, and iNKT cells appear to become diverted to a naïve-like fate, characterized by lack of expression of memory or NK markers and a failure to produce effector cytokines. Conversely when HDAC7 function is lost, naïve development is reduced, more thymocytes develop as EOMES-expressing CD8 innate effectors, and the fraction of peripheral CD4 and CD8 T cells expressing memory markers and primed for cytokine production increases. Thus, appropriately regulated nuclear export of HDAC7 appears to be a licensing step that permits both negative selection and the acquisition of alternative cell fates, such as PLZF-dependent agonist selection to the iNKT lineage.

In this study, we focused on iNKT cells due to their relatively high abundance and easy identification using CD1D tetramers, but we suspect that HDAC7-ΔP similarly abrogates development of other PLZF-dependent innate effector subtypes, including rare MR1-restricted MAIT cells and γδ NKT cells (*Chandra and Kronenberg, 2015*; *Fan and Rudensky, 2016*). In contrast, another well-described innate effector type, CD8αα + IELs localized in small intestine (*Mayans et al., 2014*), are only slightly reduced in HDAC7-ΔP mice (*Figure 1—figure supplement 1A*), consistent with their PLZF-independent derivation (*Cheroutre et al., 2011*). The identification of a committed precursor to innate lymphoid cells that transiently expresses high amounts of PLZF (*Constantinides et al.,*

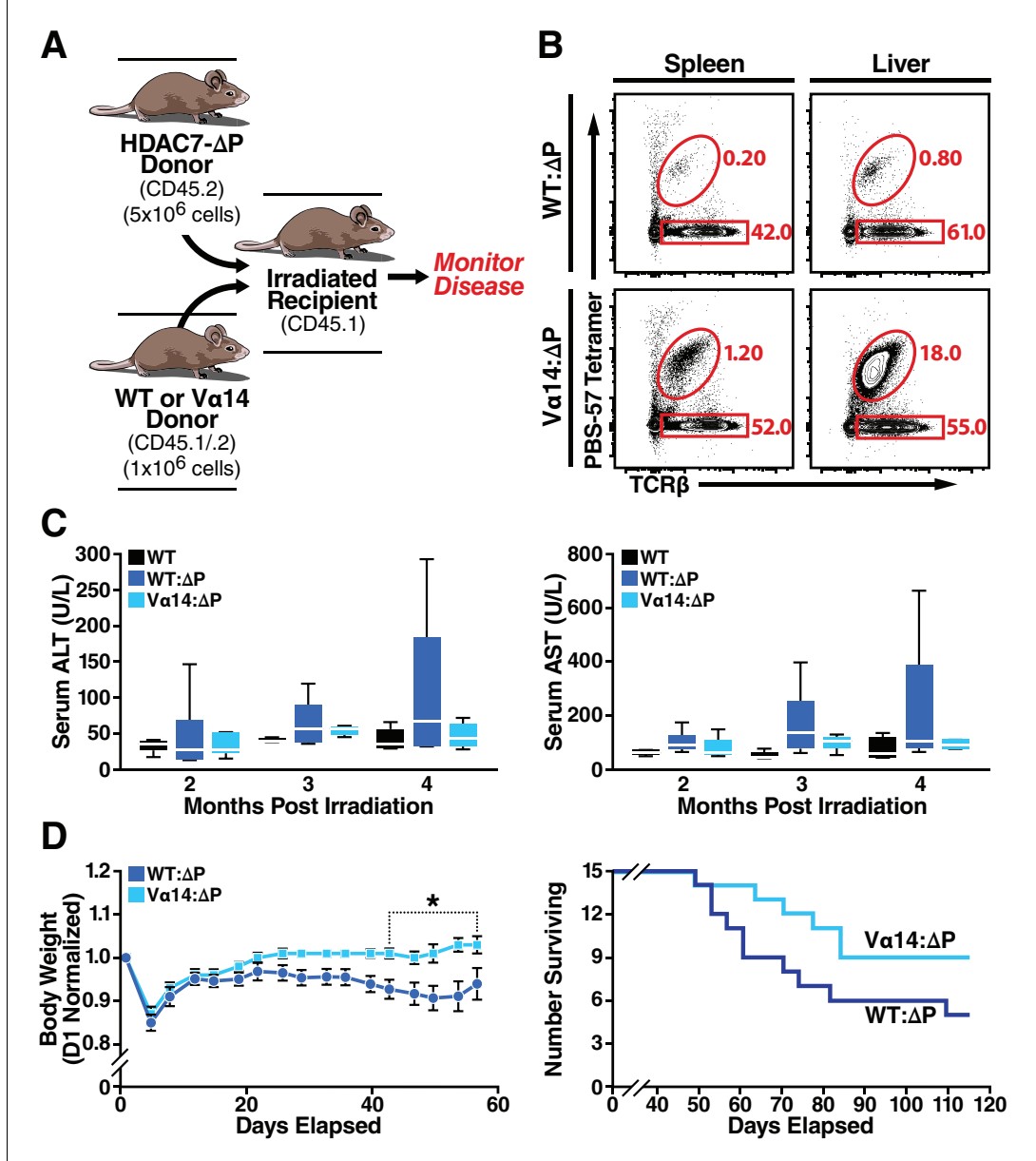

**Figure 8.** Loss of iNKT Cells in HDAC7ΔP Mice Contributes to Tissue-Specific Autoimmunity. (**A**) Schematic of mixed BM chimeras used to monitor HDAC7-ΔP-mediated autoimmunity time course and severity. Lethally irradiated CD45.1 BoyJ recipients were reconstituted (6 × 10⁶ cells) with a 1:5 mixture of either WT (CD45.1): HDAC7-ΔP (CD45.2) or Vα14 (CD45.1): HDAC7ΔP (CD45.2) bone marrow cells. (**B**) Vα14 bone marrow (bottom) robustly restores peripheral iNKT cells (Tet⁺ TCRβ⁺ in liver and spleen in mixed BM chimeras, while WT bone marrow does not. Plots are representative of two sets of independently made chimeras. (**C**) Plasma concentration of liver (ALT, AST) markers of tissue damage over time measured in WT mice compared to Vα14: HDAC7-ΔP and WT: HDAC7-ΔP BM chimeras. (**D**) Body weight (left) and survival (right) of mixed BM chimeras over time post-irradiation. Weights in (**D**) were normalized to starting weight on Day one post-irradiation and measured twice a week thereafter. Survival (D, right) was assessed by monitoring for spontaneous death twice a week or by euthanasia after reaching a clinical endpoint of at least 20% body weight loss compared to peak weight post-irradiation. Using Kaplan-Meier analysis, p=0.0616 by Gehan-Breslow-Wilcoxon tests. Bars on graphs indicate mean ±SEM (error bars); whiskers on box-and-whiskers plots represent min to max. Data in (**C**) were collected from N = 6 mice per group; data in (**D**) and (**E**) were combined from three independent experiments with N = 16 mice total per group. Statistical significance in (**D**) was determined using two-way ANOVA; *p≤0.05. Bonferroni post-tests were used for pairwise comparisons.

DOI: https://doi.org/10.7554/eLife.32109.022

The following source data and figure supplement are available for figure 8:

**Source data 1.** Microsoft Excel workbook containing numerical data matrices for all figure panels (on separate sheets) in which individual data points are not represented graphically.

*Figure 8 continued on next page*

*Figure 8 continued*

DOI: https://doi.org/10.7554/eLife.32109.024

**Figure supplement 1.** Supporting Data on restoration of iNKTs in HDAC7-ΔP mixed chimeras with Vα14-Jα18 TG bone marrow and autoimmune disease course.

DOI: https://doi.org/10.7554/eLife.32109.023

*2014*) also raises the intriguing possibility that development of these cell types may be regulated by class IIA HDACs as well. Furthermore, the main mechanism of action we investigate here, HDAC7 antagonism of PLZF via direct interaction, may be generalizable to other members of the BTB-POZ-ZF family. For example, the signature transcription factor of Tfh cells, Bcl6, is known to associate with HDAC4 (*Lemercier et al., 2002*) (*Crotty, 2014*). A class IIA HDAC/BTB-ZF axis may thus regulate T cell or ILC development at additional branch points. Additionally, in recent years a number of transcriptional regulators and epigenetic modifiers – including JARID2, NKAP, HDAC3, and EZH2 (*Pereira et al., 2014*; *Thapa et al., 2013*; *Dobenecker et al., 2015*) – have been identified that regulate iNKT ontogeny. At least one member, HDAC3, physically associates with class IIA HDACs as part of a larger co-repressive complex (*Fischle et al., 2002*). Devising systems to investigate these relationships as well as HDAC7 association with PLZF via ChiP-Seq and other genomic-scale approaches is a current priority in our laboratory.

## HDAC7 control of iNKT cell development modulates the susceptibility of liver to autoimmune attack

By restoring the missing iNKT population with the use of Vα14 donor bone marrow, we significantly attenuated the severity and time course of HDAC7-ΔP-mediated autoimmune liver disease, resulting in improved liver function, better body weight maintenance, and reduced overall mortality. Although specific rescue of iNKT cells did not provide protection in all tissues – almost all Vα14: HDAC7-ΔP chimeras eventually developed the same ultimately lethal exocrine pancreatitis as WT: HDAC7-ΔP chimeras – our studies nonetheless reveal an important new role for impaired iNKT development as an exacerbating factor in liver autoimmunity. Since both HDAC7 and PLZF influence the development of several non-iNKT innate effector subtypes that would not have been restored with Vα14 bone marrow, it is tempting to speculate that restoring these other subsets might ameliorate tissue destruction and T-cell infiltration due to HDAC7-ΔP in other organs.

Innate effector T-cells are often considered frontline first-responders to infection that amplify and orchestrate the early immune response to invading pathogens. Thus, it was somewhat surprising to uncover a protective or anti-inflammatory role for iNKT cells in attenuating tissue destruction. Additional studies will be required to uncover the mechanisms through which iNKT cells provide protection, but for now we favor a model in which innate effectors occupy tissue niches at their sites of residence, limiting access of other immune cells into those sites. Alternatively, the loss of iNKT cells in these tissues may compromise normal mucosal barrier function in a manner that promotes inflammation and the subsequent recruitment of autoreactive Tconv. In HDAC7-ΔP mice, escape of autoreactive Tconv due to impaired negative selection may thus produce a potentially but not necessarily pathogenic population, which requires the additional loss of PLZF-dependent innate effectors from their target tissues to create an opening for infiltration. This 'two-hit' model may explain multiple types of tissue-specific autoimmunity, in which genetic lesions that generate excess self-reactive lymphocytes are coupled with separate or related defects in tissue-resident innate effector populations at specific sites, rendering these tissues particularly vulnerable to attack.

Our findings likely hold considerable relevance to understanding the etiology and mechanisms contributing to some types of human autoimmunity. Indeed, common variant single nucleotide polymorphisms (SNPs) in the HDAC7 gene are significantly associated with human autoimmune and auto-inflammatory diseases, namely primary sclerosing cholangitis (*Liu et al., 2013*) and inflammatory bowel disease (*Jostins et al., 2012*). Additional common variant SNPs in kinases known to export Class IIA HDACs via phosphorylation, including SIK2 and PRKD2, are also associated with primary sclerosing cholangitis (*Liu et al., 2013*), suggesting aberrant regulation of HDAC7 nuclear export as a causative mechanism. Moreover, the genes that we identified as regulated by HDAC7 in iNKT development show a striking overlap with other risk loci from these GWAS studies (*Figure 6A*), suggesting that the broad HDAC7 regulatory network may be a crucial nexus that underlies

susceptibility to several autoimmune diseases of considerable clinical importance. Indeed, mapping the overlapping GWAS loci to their associated signaling networks revealed a remarkable clustering around a few important signaling pathways in iNKT and effector development, including IL12, IL21, IL18, IFNG, and IL4, as well as Ig- and TNF-superfamily costimulatory pathways. Deciphering the complex relationship between HDAC7, PLZF and other HDAC7 interaction partners, the observed modulation of these pathways, and the resulting cellular and pathologic phenotypes will be a major task for us going forward. We hope that this effort will illuminate the way forward in translating our finding that reestablishing missing iNKT cells can ameliorate HDAC7-mediated hepatic autoimmunity into potential therapeutic modalities for the analogous human diseases, based on the restoration of innate effector function.

# Materials and methods

**Key resources table**

| Reagent type (species) or resource | Designation | Source or reference | Identifiers | Additional information |
|---|---|---|---|---|
| Gene (*Homo sapiens*) | Histone Deacetylase 7 (HDAC7) | NA | HDAC7 | Coding sequence used for HDAC7 expression constructs and the HDAC7-ΔP transgene. |
| Gene (*Homo sapiens*) | Promyelocytic Leukemia, Zinc Finger (PLZF) | NA | ZBTB16 | Coding sequence used for PLZF expression constructs. |
| Strain, strain background (*Mus musculus*) | C57/BL6 | Jackson Laboratories | Stock Number: 000664 | |
| Strain, strain background (*Mus musculus*) | Boyj (B6.SJL-Ptprca Pepcb/BoyJ) | Jackson Laboratories; PMID: 11698303 | Stock Number: 002014 | B6 strain congenic for *Cd45.1* |
| Genetic reagent (*Mus musculus*) | Vα14/Jα18 Transgenic, Tg(Cd4-TcraDN32D3)1Aben | Jackson Laboratories; PMID: 18031695 | MGI:4880641 | Vα14/Jα18 Transgenic from Bendelac laboratory |
| Genetic reagent (*Mus musculus*) | Lck-Cre Transgenic, Tg(Lck-cre)548Jxm | Jackson Laboratories; PMID: 8618846 | MGI: 2176199 | Cre strain for thymic *Hdac7* deletion |
| Genetic reagent (*Mus musculus*) | *Hdac7^flox/flox*, Hdac7tm2Eno | Eric Olson, UTSW; PMID: 16873063 | MGI: 1891835 | HDAC7 floxed allele |
| Genetic reagent (*Mus musculus*) | *Hdac7^-/+*, Hdac7tm1Eno | Eric Olson, UTSW; PMID: 16873063 | MGI: 1891835 | HDAC7 null allele |
| Genetic reagent (*Mus musculus*) | Lck-PLZF transgenic, C57BL/6-Tg(Cd4-Zbtb16)1797Aben/J | Jackson Laboratories; PMID: 18703361 | MGI:4881493 | PLZF Transgenic strain from Bendelac Laboratory |
| Genetic reagent (*Mus musculus*) | HDAC7-ΔP Transgenic | Our laboratory; PMID: 23103766 | NA | Transgenic expression of HDAC7-ΔP under control of Lcl proximal promoter/CD2 LCR in C57BL/6 |
| Recombinant DNA reagent | pCDNA 3.1(+) FLAG-HDAC7 FL | this paper | NA | Expression construct for FL FLAG-tagged human HDAC7 |
| Recombinant DNA reagent | pCDNA 3.1(+) FLAG-HDAC7 1–497 | this paper | NA | Expression construct for FLAG-tagged human HDAC7 truncation mutant |
| Recombinant DNA reagent | pCDNA 3.1(+) FLAG-HDAC7 65–497 | this paper | NA | Expression construct for FLAG-tagged human HDAC7 truncation mutant |
| Recombinant DNA reagent | pCDNA 3.1(+) FLAG-HDAC7 120–497 | this paper | NA | Expression construct for FLAG-tagged human HDAC7 truncation mutant |
| Recombinant DNA reagent | pCDNA 3.1(+) FLAG-HDAC7 1–220 | this paper | NA | Expression construct for FLAG-tagged human HDAC7 truncation mutant |

*Continued on next page*

*Continued*

| Reagent type (species) or resource | Designation | Source or reference | Identifiers | Additional information |
|---|---|---|---|---|
| Recombinant DNA reagent | pCDNA 3.1(+) FLAG-HDAC7 1–180 | this paper | NA | Expression construct for FLAG-tagged human HDAC7 truncation mutant |
| Recombinant DNA reagent | pCDNA 3.1(+) HA-PLZF FL | this paper | NA | Expression construct for FL HA-tagged human PLZF |
| Recombinant DNA reagent | pCDNA 3.1(+) HA-PLZF 1–318 | this paper | NA | Expression construct for HA-tagged human PLZF truncation mutant |
| Recombinant DNA reagent | pCDNA 3.1(+) HA-PLZF 1–405 | this paper | NA | Expression construct for HA-tagged human PLZF truncation mutant |
| Recombinant DNA reagent | pCDNA 3.1(+) HA-PLZF 105–460 | this paper | NA | Expression construct for HA-tagged human PLZF truncation mutant |
| Recombinant DNA reagent | pCDNA 3.1(+) HA-PLZF 280–673 | this paper | NA | Expression construct for HA-tagged human PLZF truncation mutant |
| Recombinant DNA reagent | pCDNA 3.1(+) HA-PLZF 320–673 | this paper | NA | Expression construct for HA-tagged human PLZF truncation mutant |
| Recombinant DNA reagent | pCDNA 3.1(+) HA-PLZF 395–673 | this paper | NA | Expression construct for HA-tagged human PLZF truncation mutant |
| Recombinant DNA reagent | pCDNA 3.1(+) HA-PLZF 455–460 | this paper | NA | Expression construct for HA-tagged human PLZF truncation mutant |
| Recombinant DNA reagent | pCDNA 3.1(+) GAL4-PLZF FL | this paper | NA | Expression construct for GAL4 DNA-binding domain (1-142) fused to full-length PLZF |
| Recombinant DNA reagent | pCDNA 3.1(+) GAL4-PLZF 455–460 | this paper | NA | Expression construct for GAL4 DNA-binding domain (1-142) fused to PLZF1 - 460 truncation mutant |
| Recombinant DNA reagent | pCDNA 3.1(+) GAL4-PLZF 455–460 | this paper | NA | Expression construct for GAL4 DNA-binding domain (1-142) fused to PLZF1 - 318 truncation mutant |
| Software, algorithm | Ingenuity Pathway Analysis | Qiagen | RRID:SCR_008653 | Tool for pathway mapping and other gene ontology analysis from RNAseq transcript abundance data. |
| Software, algorithm | Bowtie 2.0 | Johns Hopkins University; PMID: 22388286 | RRID:SCR_005476 | Tool for aligning raw sequence reads to genome |
| Software, algorithm | SeqMonk | Babraham Institute | RRID:SCR_001913 | Tool for calculating RNA transcript abundances from Bowtie-mapped sequence reads (.SAM files) |

## Study Design

The initial objective of this work was to investigate the molecular mechanisms behind the control of iNKT development by HDAC7, which was an observation we made incidentally in our prior characterization of the general role of HDAC7 in thymic T cell development. The idea that HDAC7 might do this at least in part via interaction with PLZF arose from the review of older literature on these molecules showing they interact. The notion that the role of HDAC7 in iNKT cells has a bearing on the tissue distribution of autoimmunity due to altered HDAC7 function arose from the concordance between NKT-populated tissues and those showing disease in HDAC7-ΔP transgenic mice. This idea was highlighted in importance by the publication of GWAS studies after the initiation of our work

that statistically associated HDAC7 and its regulatory network with human diseases affecting the same tissues. We investigated these questions using a combination of cell culture and transgenic mouse models in which the function of HDAC7 and/or PLZF was altered in thymocytes. Parameters measured include cellular abundance in different tissues, T cell effector function after ex-vivo stimulation, luciferase expression, protein-protein interactions, global transcript abundance, and various clinical measures associated with autoimmune disease, as detailed in the following sections.

With the exception of our RNA-seq study, which was done in one experiment using three biological replicates for each condition, all results depicted in this work are based on at least two completely independent trials, comprising at least three biological replicates, that is data from three separate animals of each genotype or from three separate transfections of reporter/expression constructs. Larger sample sizes than this were used as feasible, based on the availability of experimental genotypes of interest, prospective estimates of the statistical power required to show significance for effects of the magnitudes initially observed, and the constraints of time and resources required for analysis. No data that were collected were excluded from the study unless there was clear evidence of a technical failure in data collection, or in the case of the animal studies, morbidity/mortality that was clearly unrelated to the pathologic conditions under study. Except where otherwise indicated in the figure legends, all control-experimental pairs were composed of sex-matched littermates, and all primary immune phenotypes were measured in animals between 4 and 8 weeks of age.

## Mouse Strains and procedures

All mice were housed in a specific pathogen-free barrier facility at the Gladstone Institutes, in compliance with NIH guidelines and a UCSF IACUC animal use protocol. All experimental strains were on a C57BL/6 (B6) genetic background. B6, BoyJ, V$\alpha$14/J$\alpha$18 transgenic (Tg(Cd4-TcraDN32D3) 1Aben) and PLZF transgenic (C57BL/6-Tg(Cd4-Zbtb16)1797Aben/J) mice were obtained from The Jackson Laboratory, Bar Harbor, ME. Mice deleting *Hdac7* (*Hdac7*$^{flox:-}$::*lck*$^{cre}$) or expressing the HDAC7-$\Delta$P transgene under the control of the *Lck* proximal promoter were prepared as described elsewhere (*Kasler et al., 2011*) (*Kasler et al., 2012*). Hematopoietic chimeras were prepared as follows: Recipients (8–10 wk-old BoyJ or BoyJ X B6) mice were irradiated with a split dose of 700 + 500 Rads, 3 hr. apart, from a $^{137}$Cs source (J.L. Shepherd and Associates, San Fernando, CA). Mice were reconstituted with 5 $\times$ 10$^6$ bone marrow cells from WT (Boyj or B6 X BoyJ heterozygote), HDAC7-$\Delta$ P TG (CD45.2), *Hdac7-KO* (CD45.2), or V$\alpha$14/J$\alpha$18 (CD45.2) transgenic donors, injected retro-orbitally in 200 µl of PBS. Bone marrow cell suspensions were prepared by crushing tibias and femurs, dissociating marrow cells in PBS, and purifying mononuclear cells by Ficoll gradient centrifugation. Serum for AST/ALT/lipase analysis was collected by tail vein incision and analyzed by the UCSF Clinical Laboratory at SFGH.

## Flow cytometric analysis of immune cell populations

Cell suspensions were prepared from mouse thymus and spleen by crushing, dissociation of cells by pipetting, straining through 40 µm nylon mesh, and ACK lysis. Magnetic enrichment of iNKT cells from ~2×107 thymocytes was performed using APC-conjugated PBS-57 tetramers with the Easy-Sep (StemCell Technologies, Cambridge, MA) APC Positive Selection Kit, according to the package directions. Lymphocytes were prepared from liver by mincing of the tissue, straining through a 40 µm nylon mesh, and discontinuous Percoll gradient centrifugation. Intestinal intra-epithelial lymphocytes were prepared by extensive flushing of whole small intestines with cold PBS, excision of Peyer's patches under magnification using a Trypan Blue-filled pipet as contrast medium, cutting into ~5 mm longitudinally opened segments, and incubation at 37°C with rocking in PBS with 2 mM DTT and 5 mM EDTA for 30 min. IEL were then further purified from the dissociated epithelium by Percoll gradient centrifugation. For analysis of cytokine expression, cells were cultured for 4 hr post-isolation with 50 ng/ml PMA (MilliporeSigma, St. Louis, MO) plus 0.5 µM ionomycin (MIlliporeSigma) and for 1 hr with 0.5 µg/ml Brefeldin A (MilliporeSigma) prior to staining. Viability staining was performed for 15 min in the dark at room temperature using eFluor 520 or eFluor 780 fixable viability dyes (Thermo Fisher Scientific, Waltham, MA) at 1:1000 in PBS. Surface staining with CD1D tetramers and fluorochrome-conjugated antibodies was performed for 30 min on ice in PBS with 2% FCS, followed by either fixation in PBS/1% PFA or fixation/permeabilization with the eBioscience

FOXP3 intracellular staining kit (Thermo Fisher). Intracellular staining for cytokines or transcription factors was performed for 1 hr on ice in eBioscience FOXP3 Perm/wash buffer. Analytical flow cytometry was performed using a BD (Becton Dickinson, Franklin Lakes, NJ) LSRII Cytek (Cytek Bioscinces, Fremont, CA) FACS Calibur DxP8 instrument. Data processing for presentation was done using FlowJo 10.0 (BD). Cell sorting was performed using a BD FACS-Aria II instrument. CD1D-α GalCer tetramers (PBS-57), conjugated with either phycoerythrin (PE) or allophycocyanin (APC) were obtained from the NIH tetramer core (http://tetramer.yerkes.emory.edu/). The following commercial antibodies were used for flow cytometry: CD11a-PE-Cy7 (Thermo Fisher), clone M17/4; CD18-PE (Thermo Fisher), clone M18/2; CD24-PE-Cy7 (BD), clone M1/69; CD3-APC-EF780 (Thermo Fisher), clone 2C11; CD4-BV650 (BD), clone RM4-5; CD4-PE (BD), clone GK1.5; CD4-APC (BD), clone RM4-5; CD44-PE-Cy7 (Thermo Fisher), clone IM7; CD44-APC-Cy7 (BD), clone IM7; CD44-APC (Thermo Fisher, clone IM7; CD45.1-Pacific Blue (Thermo Fisher), clone A20; CD45.1-FITC (Thermo Fisher), clone A20; CD45.2-V500 (BD), clone 104; CD45.2-PE-Cy7 (BioLegend, San Diego, CA), clone 104; CD5-APC (BD), clone 53–7.3; CD62L-APC-Cy7 (BD), clone MEL-14; CD69-PE (Thermo Fisher), clone H1 2F3; CD8-Alexa 700 (Tonbo Biosciences, San Diego, CA), clone 53–6.7; CD8-PerCP (BioLegend), clone 53–6.7; CXCR3-PE (Thermo Fisher), clone cxcr3-173; Eomes-PE (Thermo Fisher), clone Dan11-mag; Ly6C-APC (Thermo Fisher), clone hk1.4; NK1.1-PE-Cy7 (BD), clone pk136; NK1.1-APC-Cy7 (BD), clone pk136; PLZF-PE (Thermo Fisher), clone Mags.21f7; T-bet-PE-Cy7 (Thermo Fisher), clone ebio4b10; TCRβ-PerCP-5.5 (BD), clone H57-597; TCRβ-APC-Cy7 (BD), clone h57-597; TCRγδ-APC (BD), clone GL3; Vg6.3/6.2-PE (BD), clone 8f4h7b7.

## RNA-seq analysis of gene expression

Cell suspensions were prepared from thymus and spleen of 6–8 week old wild type B6, Vα14/Jα18 transgenic, or Vα14/Jα18 X HDAC7-ΔP mice. iNKT cells were sorted by FACS using antibodies to TCRβ(+) and the PBS-57 CD1D-αGalCer tetramer(+). Naïve Tconv were sorted using antibodies to CD4(+), CD8(-), TCRβ(+), and CD44(-). Cells (250,000–2,000,000) were purified from three littermate triplets for each strain (18 samples total), and total RNA (200 ng to 4 μg) was prepared using the Rneasy Plus Mini Kit (Qiagen inc., Venlo, The Netherlands). Double-stranded cDNA libraries were prepared by the Gladstone Institutes Genomics Core using the Nugen Ovation kit (Nugen, San Carlos, CA). The Libraries were sequenced by the UCSF Center for Advanced Technology using the Illumina HiSeq 4000 instrument (Illumina, San Diego, CA). Six barcoded samples were loaded per lane. FASTQ files (approximately $5.5 \times 10^7$ reads each) were mapped to the UCSC Mouse genome Build 37 (Mm.9) using Bowtie2 (Johns Hopkins University). Approximately $4 \times 10^7$ (~75%) of reads per sample were mapped uniquely to the mouse genome. Gene-level tabulation, quality control, and expression analysis was done on. SAM format files generated by BOWTIE2 using SeqMonk 0.33 (http://www.bioinformatics.babraham.ac.uk/projects). Ontologic analysis and pathway mapping were performed using Ingenuity Pathway Analysis (http://www.ingenuity.com/). All primary data associated with these experiments have been deposited at GEO (https://www.ncbi.nlm.nih.gov/geo, accession GSE105026), and a summary of all gene expression data and statistics for differentially expressed genes is provided in *Supplementary file 1*.

## Plasmids, transfections, and reporter assays

The human PLZF coding sequence (RCAS(B)-Flag-PLZF), deposited by Peter Vogt (*Shi and Vogt, 2009*) was obtained from Addgene. N-terminally HA-tagged full-length PLZF was amplified from this coding sequence using the following Primers: N-terminal PLZF Bam HI, HA tag, *Eco* RV, *Bsa* BI, Hpa I: 5' aaaaaaggatccacc atg tat ccc tac gat gtt cca gat tat gcg ata tca atc gtt aac atg gat ctg aca aaa atg gg; C-terminal *Swa* 1, stop, *Not* 1: 5' cct cta cct gtg cta tgt gtg att taa atgattaga-taagcggccgcaaaaaa 3'. This amplification product was subcloned into pCDNA3.1(+) (Thermo Fisher) using *Bam* H1 and *Not* one sites. Different PLZF truncations were amplified from this construct and sub-cloned into the introduced flanking sites (further details on request). For the GAL4 DNA-binding domain-PLZF fusion constructs, the GAL4 DNA-binding sequence was amplified from a plasmid template and inserted into the *Eco* RV sites of full-length or truncated PLZF expression constructs described above. Construction of full-length human HDAC7 and HDAC7-VP16 fusion-encoding expression vectors is described elsewhere (*Dequiedt et al., 2003*). Other truncated, FLAG-tagged HDAC7 constructs were amplified from these templates and re-ligated into pCDNA3.1(+). The

GAL4 UAS(5) SV40-Firefly luciferase reporter construct was prepared by ligation of an oligonucleotide cassette containing 5 GAL4 recognition sites into the *Sma* 1 site of pGL2 Promoter (Promega Corp., Fitchburg, WI).

For pulldown experiments, 10 cm dishes seeded the previous day with $3.2 \times 10^6$ HEK 293 T cells were transfected with 20 µg of total DNA, consisting of 10 µg each of PLZF and HDAC7 constructs or the corresponding empty vectors, using $CaPO_4$/chloroquine. HEK-293T cells, were originally obtained from ATCC, and had been confirmed by PCR testing to be mycoplasma-free within 6 months of their use for these experiments. After 48 hr, cells were harvested for interaction analysis. For reporter assays, 6-well dishes seeded with $0.8 \times 10^6$ HEK 293 T cells/well were transfected using $CaPO_4$/chloroquine with 6.1 µg of total DNA, consisting of 2 µg each of gal4(5) luc, gal4-PLZF fusion construct, and empty vector or HDAC7 expression construct, plus 100 ng of EF1α *Renilla* luciferase. Cells were harvested for luciferase assay 48 hr after transfection, and luciferase activity was measured using the Promega Dual-Luciferase assay kit.

## Co-immunoprecipitations and western blots

For the co-immunoprecipitation of endogenous HDAC7 with transgenic PLZF in thymocytes, thymocyte lysates from wild-type and PLZF transgenic mice were prepared using p300 lysis buffer (250 mM NaCl, 0.1% NP-40, 20 mM NaH2PO4, pH 7.5, 5 mM EDTA, 30 mM sodium pyrophosphate, 10 mM NaF, and HALT protease/phosphatase inhibitors (Thermo Fisher). After clarification (5 min, 13,000Xg) and pre-clearing (3 hr at 4°C with proteinA/G agarose beads), lysates were immunoprecipitated with either 1 µg/ml of α-HDAC7 antibody (H-273, Santa Cruz Biotechnology, Dallas, TX) or 1 µg/ml of rabbit IgG isotype control antibody (Cell Signaling Technologies, Danvers, MA) at 4°C overnight. The lysates were then incubated with 50 µl of protein A/G agarose beads (Santa Cruz Biotechnology) at 4°C for 4 hr, followed by washing five times with p300 lysis buffer. Immunoprecipitated proteins from the beads were eluted with non-reducing Laemmli SDS PAGE sample buffer by boiling for 3 min. For pulldown analysis of HDAC7-PLZF truncation mutants, 10 cm dished were harvested and lysed in 0.8 mL of P300 buffer, clarified by spinning 5 min. at 13,000 g, then incubated for 4 hr at 4°C with 30 µL/sample of FLAG M2-agarose beads (MilliporeSigma). After four washes with p300 buffer, bound proteins were eluted from the beads by addition of 100 µL of reducing Laemmli SDS-PAGE sample buffer, followed by a 5 min incubation at 95°C.

After SDS PAGE and transfer to nitrocellulose, membranes were probed with antibodies against HDAC7 (H-273, Santa Cruz Biotechnology), PLZF (D9, Santa Cruz Biotechnology), and β-actin (Abcam, Cambridge, MA), HA epitope (Cell Signaling), or FLAG epitope (MilliporeSigma), overnight at 4°C. After washing and incubation with HRP- or IRDye-conjugated antibodies (Li-Cor Biotechnology, Lincoln, NE), signal was detected using chemiluminescence and film or a Li-Cor Odyssey scanner respectively. Bands for quantitative pulldown analysis were quantified from the scanner output using ImageJ (Wayne Rasband, National Insititutes of Health)

## Acknowledgements

We thank G Maki and T Roberts for figure preparation, Ethan Pak, M Cavrois, M Maiti, A Uebersohn for technical assistance, C Doherty for coordinating plasma lipase and liver panel measurements, A Abbas, A Chawla, D Sheppard, and members of the Verdin laboratory for helpful comments and discussion, and M Ott and J Roose for critically reading the manuscript.

## Additional information

### Competing interests

Hyung W Lim: is currently affiliated with Novartis Institutes for Biomedical Research (NIBR), but the research was conducted when he was at the Gladstone Institute/University of California. The author has no competing financial interests to declare. The other authors declare that no competing interests exist.

## Funding

| Funder | Grant reference number | Author |
|---|---|---|
| National Institutes of Health | AI117864 | Eric Verdin |
| Kurtzig and Mulholland Families | | Eric Verdin |
| Gladstone Institutes | | Eric Verdin |
| National Institutes of Health | DA041742 | Eric Verdin |

The funders had no role in study design, data collection and interpretation, or the decision to submit the work for publication.

## Author contributions

Herbert G Kasler, Conceptualization, Formal analysis, Validation, Investigation, Visualization, Methodology, Writing—original draft, Writing—review and editing; Intelly S Lee, Conceptualization, Formal analysis, Validation, Investigation, Visualization, Methodology, Writing—original draft; Hyung W Lim, Formal analysis, Investigation, Visualization, Methodology; Eric Verdin, Conceptualization, Resources, Supervision, Funding acquisition, Methodology, Project administration, Writing—review and editing

## Author ORCIDs

Herbert G Kasler (iD) http://orcid.org/0000-0002-6660-4267
Eric Verdin (iD) http://orcid.org/0000-0003-3703-3183

## Ethics

Animal experimentation: All mice were housed in specific pathogen-free barrier facilities at the Gladstone Institutes or the Buck institute. All animal care and animal experiments were carried out in compliance with NIH guidelines and IACUC-approved UCSF (AN110172) or Buck Institute (A10154) animal use protocols.

## Decision letter and Author response

Decision letter https://doi.org/10.7554/eLife.32109.036
Author response https://doi.org/10.7554/eLife.32109.037

---

# Additional files

## Supplementary files

• Supplementary file 1. Excel spreadsheet containing SeqMonk Normalized expression values for all present RNAs in our 18 samples (six genotypes, three biological replicates each, as defined in Materials and methods), with means for each genotype (Columns A-Z), summary statistics for key comparisons (mean, $\log_2$ mean/mean, and T-test, Columns AA-AK), and aligned data from relevant published studies (Columns AL-AT). Additional notes and PMIDs for gene-specific published findings for disease-associated GWAS loci are provided in Columns AY-BA.
DOI: https://doi.org/10.7554/eLife.32109.025

• Supplementary file 2. Full table of Ingenuity Pathway Analysis overrepresented pathways for the comparison of genes expressed in CD4 SP cells for Vα14Jα18 TG X HDAC7-ΔP TG mice vs Vα14Jα18 TG littermates in spleen and thymus.
DOI: https://doi.org/10.7554/eLife.32109.026

• Supplementary file 3. Full table of Ingenuity Pathway Analysis predicted upstream regulators and their targets for the comparison of genes expressed in CD4 SP cells for Vα14Jα18 TG X HDAC7-ΔP TG mice vs Vα14Jα18 TG littermates in spleen and thymus.
DOI: https://doi.org/10.7554/eLife.32109.027

• Transparent reporting form
DOI: https://doi.org/10.7554/eLife.32109.028

## Major datasets

The following dataset was generated:

| Author(s) | Year | Dataset title | Dataset URL | Database, license, and accessibility information |
|---|---|---|---|---|
| Kasler HG, Lee IS, Lim HW, Verdin E | 2018 | Gene regulation by Histone Deacetylase 7 during invariant Natural Killer T Cell development | https://www.ncbi.nlm.nih.gov/geo/query/acc.cgi?acc=GSE105026 | Publicly available at the NCBI Gene Expression Omnibus (accession no: GSE105026). |

The following previously published datasets were used:

| Author(s) | Year | Dataset title | Dataset URL | Database, license, and accessibility information |
|---|---|---|---|---|
| Mao AP, Constantinides MG, Mathew R, Zuo Z, Bendelac A | 2016 | Multiple levels of transcriptional regulation by PLZF in NKT cell development | https://www.ncbi.nlm.nih.gov/geo/query/acc.cgi?acc=GSE81772 | Publicly available at the NCBI Gene Expression Omnibus (accession no: GSE81772) |
| Yang L | 2009 | Immunological Genome Project data Phase 1 | https://www.ncbi.nlm.nih.gov/geo/query/acc.cgi?acc=GSE15907 | Publicly available at the NCBI Gene Expression Omnibus (accession no: GSE15907) |

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
