## [Decision Letter]

Thank you for submitting your article "HDAC7 Mediates Tissue-Specific Autoimmunity via Control of Innate Effector Function in Invariant Natural Killer T-Cells" for consideration by *eLife*. Your article has been reviewed by three peer reviewers, and the evaluation has been overseen by a Reviewing Editor and Michel Nussenzweig as the Senior Editor. The reviewers have opted to remain anonymous.

The reviewers have discussed the reviews with one another and the Reviewing Editor has drafted this decision to help you prepare a revised submission.

Summary:

This is an interesting paper that demonstrates an important role for HDAC7 in NKT cell development and function.

Essential revisions:

Although the manuscript had interesting information, there were a number of concerns that need to be addressed in a revised manuscript. Among the major issues (detailed in the complete reviews attached below) were:

1) There were concerns about controls and data not shown. The authors should:a) Show control CD1d tetramer staining data for all relevant figures;b) For rare populations, it is critical to show the entire gating strategy (and confirm whether they performed doublet exclusion). This should be shown in the supplemental data;c) Data not shown should be included in the supplemental data.

2) Reviewers were concerned about the low level of NK1.1 expression in control animals that seemed unusual.

3) Did g/d T cells compensate for IL4 expression and phenotype of innate CD8 T cells?

4) Authors need to pull back from statements in the Results section that they do not test experimentally. They do not test why there are fewer iNKT cells, especially at later stages. Is it proliferation? increased cell death? Block in thymic effector differentiation (NKT1/NKT2/NKT17 by examining Tbet/Gata3/RORgt)? If they will speculate without experimentation it should be moved to the Discussion rather than the Results.

5) Acknowledgement that their gene array doesn't compare the same populations between the WT and KO animals.

Again, the full text of all reviews are given below. Please note that a revised manuscript should address all of the concerns cited in the full reviews, though the major issues are listed above and should be addressed separately in responses to the reviewers.

*Reviewer #1:*

In this manuscript Kasler et al. report a role for the histone deacetylase HDAC7 in controlling the development of Natural Killer T cells, through its regulation of PLZF expression and transcriptional activity. They demonstrate a direct binding between HDAC7 and PLZF that affects PLZF transcriptional activity. Finally, through gene network analysis, they provide some tenuous link between target genes that have been linked in the autoimmunity that develops upon alteration of HDAC7 expression and genes that are affected by HDAC7 expression in iNKT cell development. They further test this idea by showing that restoring iNKT cells moderately ameliorates the tissue-specific autoimmunity that develop in HDAC7ΔP mice.

This is an interesting paper that clearly demonstrates an important role for HDAC7 in NKT cell development and function. My comments (below) are mostly technical and while they should be addressed and might slightly change the narrative of the manuscript, they will not change the overall message.

In Figure 1 (as well as several other figures in the manuscript), the various gates used for the analysis by flow cytometry seem to be placed freely. For example, the gate used to define the CD1d tetramer+ cells often incorporates cells that are TCR negative (Figure 1, Figure 3B, Figure 4). It would have been useful to stain with the control CD1d tetramer to ensure that the gated events are real NKT cells. While this reviewer agrees that it does not change the overall results (that NKT cells are affected by HDAC7), it can change the interpretation of certain results. For example, very few events corresponding to NKT cells are left in the HDAC7ΔP mice. By gating these events, it is concluded that the cells are very immature (CD44-, CD24+, NK1.1-) and blocked in their development. It would be worthwhile to verify that these very few events are real NKT cells.

Stage 0 NKT cells represent 300-1000 cells per thymus and are usually only visualized after magnetic bead enrichment of tetramer+ cells. These experiments were not carried out in that fashion here and the CD24+ cells would benefit from the examination of other markers to make sure that they represent the cells that the authors think they examine. To this end, stage 0 NKT cells are also CD69+ and Egr2+. Similarly, it is unclear to this reviewer why only 9% of the NKT cells in wildtype mice express NK1.1 (Figure 1C), while in B6 mice, the vast majority of these cells are in fact NK1.1+ (as seen in Figure 3C for example). Were all experiments performed with mice of the same age? Is it a parameter that matters?

In Figure 2, it is unclear why Figure 2B shows staining in the spleen while the other panels of the figures pertain to the thymus. Similarly to the previous comment, it is unclear why only 30% of the cells would be expressing NK1.1 in that tissue.

The development of "innate" CD8 T cells that are CD44+, CD122+, Eomes+ was reported to be dependent upon IL-4 signaling in the thymus. The source of IL-4 in the thymus is thought to be NKT2 and PLZF+ gd T cells. The presented experiments clearly demonstrate that the increased proportion of CD8^+^ CD44+ Eomes+ in HDAC7ΔP mice is decoupled from an increased proportion of NKT2 cells. Did the author examine whether this was compensated by an increase proportion of gd PLZF+ T cells? (Figure 4 would argue that it is not although it was not formally tested). The author should also modify the text of the manuscript in that the "innate" programs that might be regulated by HDAC7 are not the same in CD8 and NKT cells. NKT cells do not express Eomes and CD8 T cells do not express PLZF so the statement that "loss of HDAC7 results in the aberrant adoption of NKT-like innate effector programming" is not correct.

NKT cell can adopt different fate in the thymus (NKT1, NKT2 and NKT17) that can be visualized by the expression of the various master transcription factors associated with the phenotype. It would have been interesting to stain NKT cells for Tbet, Rorgt in addition to PLZF. The expression of PLZF protein is essentially gone in HDAC7ΔP mice (Figure 4A), yet analysis of mRNA expression (Figure 5A) reveals a very modest loss of Zbtb16 mRNA expression. How do the authors reconcile these findings? Does HDAC7 binding to PLZF (Figure 7) leads to its degradation in vivo?

It is also unclear why gene expression in total NKT cells was examined in Figure 5, while it is clear from Figure 1 that the cells did not reach the same stage of development between WT and HDAC7ΔP mice. Thereby, finding differences in gene expression for genes that are acquired late in the development of NKT cells (NK1.1, Ly49, Tbet) is perhaps not surprising, but might be unrelated to HDAC7 direct activity. It would have been more interesting, perhaps, to test differential gene expression in CD44neg NK1.1- NKT cells between the two strains.

*Reviewer #2:*

Kasler et al. analyzed the effects of Hdac7 gain of function or deletion on the immune system in mice. They tell an interesting story and weave their data with complementary studies of human autoimmunity arising from polymorphism of Hdac7 and its regulatory kinases. This makes for compelling reading and integration of current knowledge across several fields.

The paper begins with analysis of the effects of Hdac7 gain-of-function (GOF) or thymic deletion and basically shows that GOF leads to failure of iNKT development, whereas thymic deletion leads to certain thymic CD8 T cells developing features of innate effector (CD44hiEomes+) cells. The authors then bridge to consideration of Hdac7 as a negative regulator of the transcription factor, PLZF, typically stably expressed by iNKT cells.

The work has considerable rigor and interest, and is a sound and well performed study integrating aspects of molecular and cellular immunology.

*Reviewer #3:*

The main conclusion is that HDAC7 is critical for controlling whether T cells act like conventional naïve T cells or quickly reactive innate-type lymphocytes. However, there are several concerns and potential alternative interpretations.

Data is not shown regarding the conventional CD4 compartment in the periphery. They state "We observe a much more modest degree of abnormality in the CD4 compartment (data not shown)". This data needs to be added to the Results and is necessary to assess their conclusions. If HDAC7ΔP does not alter the conv T cell compartment into a more innate like phenotype, does this alter the conclusions drawn?

Previously, the authors demonstrated that use of the HDAC7ΔP transgene inhibited negative selection of autoreactive thymocytes and that these mice developed lethal autoimmunity. Is the enhanced activation of T cells observed due to activation of autoreactive cells that otherwise would have been deleted? If these are autoreactive cells, then are they more innate like or the issue is that they are being activated and are thus not naïve? If the T cells with enhanced function are ones that would have been negatively selected, then is the primary cause the block in negative selection?

Another difficulty with this manuscript are conclusions 'consistent with the data', or 'suggesting that' without doing the experiments to demonstrate the mechanism. The authors interpret the failure of Va14 tg HDAC7ΔP NKT cells to produce cytokines as 'suggesting that the cells had failed to undergo effector programming in the thymus'. This should be examined by staining for NKT1/NKT2/NKT17 using Tbet/Gata3/RORγt. Total NKT are examined for cytokine production, but if there is a defect in differentiation of the NKT subsets, then each subset should be examined individually to determine whether (for example) there are few NKT1 cells that produce normal IFNγ or many NKT1 that fail to make IFNγ. In addition, the authors state that "HDAC7ΔP blocked the intrathymic proliferation that is normally associated with post-positive selection iNKT differentiation". No experiments were performed to examine NKT proliferation after positive selection (e.g. BrdU). In addition, the loss of NKT cell at this stage could also be due another reason such as enhanced apoptosis, but this was not examined either.

"modest suppression of Treg and CD8aa IELs (data not shown)" should be added to the manuscript. CD8aa IELs are innate-like lymphocytes. If HDAC7ΔP expression leads to enhancement of innate functions, then why is this population decreased?

It is not clear that the HDAC7ΔP transgene is completely off in the periphery. In their prior manuscript originally describing this mouse, there is enhanced HDAC7 expression in peripheral T cells from HDAC7ΔP transgenic mice as compared to WT. Therefore, I would argue that their previous paper demonstrated that this transgene is not completed turned off in the periphery.

HDAC7ΔP mice have increased mature SP thymocytes – one common cause for an increase in mature SP thymocytes is a defect in thymic egress. Is there a defect in thymic egress in these mice? (KLF2, S1P1/receptor, CD69)

It is difficult to analyze the RNA-seq data presented, as the cells being examined are completely different: Vα14 (WT) NKT cells that have primarily differentiated into NKT1/NKT2/NKT17 effectors and Vα14 HDAC7ΔP NKTs that may not have differentiated into NKT1/NKT2/NKT17 effectors at all.

Empty tetramer control needs to be added to several figures, as it is not clear whether the signal observed with Tetramer/TCRβ (e.g. Figure 1) in HDAC7ΔP is above that of an empty tetramer control. If not, then further examination of this population by CD24/CD44/NK1.1 may be misleading, and the population should be examined to determine whether it expresses the canonical Vα14-Jα18 rearranged TCRα chain.

---

## [Author Response]

Essential revisions:Although the manuscript had interesting information, there were a number of concerns that need to be addressed in a revised manuscript. Among the major issues (detailed in the complete reviews attached below) were:1) There were concerns about controls and data not shown. The authors should:a) Show control CD1d tetramer staining data for all relevant figures;

The key concern seems to be that the lack of empty tetramer staining, combined with an overly inclusive gate for iNKT cells (particularly in the old Figure 1), suggests that the small number of gated events analyzed further in the HDAC7-ΔP mice (old Figure 1C) are not really iNKT cells, but rather background TCR-negative cells. We have taken several steps to address this concern. First, since we did not perform parallel empty tetramer staining in all our experiments, including the experiment used as representative data for Figure 1A in the original submission, we have replaced these panels with data from a set of animals in which we did use empty tetramer for thymus, spleen and liver (new Figure 1A, Figure 1—figure supplement 1A, Figure 2A). In doing so, we have also drawn tighter gates around the iNKT population and performed more careful spectral compensation to better distinguish and exclude TCR-negative cells.

Secondly, while the data processed in this fashion do consistently show more cells in the iNKT gate in HDAC7-ΔP thymus with loaded vs. empty tetramer (new Figure 1—figure supplement 1C) this background is still sufficiently high that it could obscure some of the findings. To try to address this issue more effectively, we performed new experiments in which we used magnetic beads to enrich tetramer-binding cells before analyzing CD44 and NK1.1 expression (as suggested by reviewer #1), and these data now replace the original Figure 1C (new Figure 1C-D). We performed similar experiments, employing empty tetramer and magnetic enrichment, for the analysis of HDAC7 KO thymocytes (new Figure 2A), which is the only other condition in which we are analyzing rare iNKT populations in this manner. Our improved analysis and these new data have caused us to modify our conclusion that there is a Stage 1 block in HDAC7-ΔP mice, as we can now detect cells at both stages 2 and 3 in the HDAC7- ΔP mice, both with magnetic enrichment (Figure 1C) and without (Figure 1, S1C, D). We have also found that the remaining cells in Stage 0-1 are in fact mostly CD24^hi^. Our original analysis was compromised by weak reagents and poor choices of color to detect CD24 and NK1.1, which made appropriate spectral compensation of our complex panel and the distinction of CD24^hi^ vs. CD24^lo^ cells difficult. While Vα14 x HDAC7-ΔP TG iNKT cells do appear to mature fairly efficiently to Stage 1 but not beyond it (new Figure 3—figure supplement 1D), our new data suggest that even this step is impaired in mice expressing only HDAC7-ΔP (New Figure 1C, D, Figure 1—figure supplement 1C-D). Additionally, since the use of contour plots to represent the distribution of the small numbers of events in these analyses was potentially misleading, we have switched to more clearly interpretable dot plots in all such cases.

b) For rare populations, it is critical to show the entire gating strategy (and confirm whether they performed doublet exclusion). This should be shown in the supplemental data;

We have now provided panels illustrating the full gating strategy for our representative tetramer staining of tissues from WT/HDAC7-ΔP mice in Figure 1—figure supplement 3. Identification of lymphocytes, doublet exclusion, and identification of live cells was performed as shown in these panels for all of our flow data.

c) Data not shown should be included in the supplemental data.

The data in question included data on the effect of HDAC-ΔP in Treg development and on the development of CD8αα IELs, which in both cases we characterized as minor in contrast to the categorical effects we observed on iNKT development in the present work and previously on negative selection (Kasler et al., 2012). With regard to the effects on Treg development, we did publish supplementary data supporting this finding in our previous paper (Kasler et al., 2012, specifically in Figures S1E and S2A). These data show an approximately 2-fold reduction in the prevalence of Treg in the HDAC7-ΔP transgenics, both in the intact mice and in mixed chimeras. When isolated, these cells showed normal activity in *ex-vivo* suppression assays. The data we obtained on CD8 αα IEL, showing a reduction in prevalence of ~2-fold in the small intestinal epithelium, have not been published previously, so we have included representative panels showing our analysis in Figure 1—figure supplement 1A. We have also now included the data for memory markers and cytokine expression in the CD4 compartment in our mixed hematopoietic chimeras (Figure 2—figure supplement 2). While the phenotypes we saw were of much lower magnitude (~30% vs. a 3-fold increase) and more variable than what we saw for the CD8 compartment, they were nonetheless statistically significant at n=6 for IL4 secretion and n=8 for memory marker expression.

2) Reviewers were concerned about the low level of NK1.1 expression in control animals that seemed unusual.

We agree with the reviewers that the profiles of CD44/NK1.1 expression for WT mice in those panels don’t make sense, and we should have caught this inconsistency before sending the manuscript out. We have since re-done this analysis for these strains with better reagents for CD24 and NK1.1, empty tetramer as a control, magnetic enrichment of iNKT cells, and more careful gating and compensation. In the new data (Figure 1C, D., Figure 1—figure supplement 1C, D), we can clearly see that WT thymic iNKT cells are predominantly at Stage 3, and that cells at stages 1-3 are all severely reduced but still detectable in HDAC7-ΔP mice. With respect to NK1.1 expression Figure 2B, those data were mislabeled and were actually from liver rather than spleen, and moreover the signal for NK1.1 was weak and had not been correctly compensated. We have substituted new data from thymus for these panels in Figure 2, with CD44/NK1.1 staging for magnetically enriched iNKT cells now provided as dot plots. The original data from this location were re-analyzed and moved to Figure 3—figure supplement 1A. We still only see about half of the iNKT cells in liver as belonging to Stage 3, however this is consistent with what we have seen in spleen and liver in other instances with the younger (i.e. 4-5 weeks old) animals that we have analyzed.

3) Did g/d T cells compensate for IL4 expression and phenotype of innate CD8 T cells?

We investigated this possibility by staining thymi from WT and HDAC7 KO mice with antibodies to the αβ and γδ TCRs, in combination with antibodies to either PLZF and the PLZF-associated Vγ6.3 TCR chain. These new analyses have been added as Figure 2—figure supplement 1A and C. We found that the overall proportion of γδ T cells was quite variable in HDAC7 KO mice, ranging from identical to WT to as much as 2.5-fold higher, while the proportion of PLZF and Vγ6.3-expressing cells within this population ranged from normal to ~2-fold reduced. Their overall prevalence thus ranged from normal to as much as 2.2-fold higher in one of six littermate pairs we analyzed. Due to this variability, the overall trend of increased representation of these cell types only reached statistical significance (P = 0.043 for one trial and P = 0.099 for another, see Figure 2—source data 1) for total γδ T cells and not Vγ6.3 or PLZF-expressing cells. We therefore do not believe it could be the cause of the expanded EOMES-expressing population we observe, as this was noted in all of the seven WT, HDAC7-KO littermate pairs we looked at.

4) Authors need to pull back from statements in the Results section that they do not test experimentally. They do not test why there are fewer iNKT cells, especially at later stages. Is it proliferation? increased cell death? Block in thymic effector differentiation (NKT1/NKT2/NKT17 by examining Tbet/Gata3/RORgt)? If they will speculate without experimentation it should be moved to the Discussion rather than the Results.

We acknowledge that we did not formally test various alternative hypotheses about what causes the defect in the numbers of iNKT cells in the HDAC7-ΔP mice, and have accordingly toned down various potentially overreaching statements in the Results section, e.g. subsection “Alteration of HDAC7 Function Dysregulates Thymic Innate Effector Programming and Interferes With iNKT Development”, first and second paragraphs; subsection “HDAC7 Regulates the Effector Programming of NKT Cells in a Manner That Mirrors the Function of PLZF”, first paragraph etc. While we still believe that HDAC7-ΔP does cause a block in effector differentiation iNKT cells, as is supported by our flow data with the Vα14 transgenics and our RNAseq analysis (note the suppression of multiple effector differentiation-associated genes in Figure 5A, including T-bet and GATA3), the limited time allowed for these revisions and the limited number of mice we had available made definitively ruling out other models with BrdU/apoptosis experiments an endeavor beyond the scope of this manuscript.

5) Acknowledgement that their gene array doesn't compare the same populations between the WT and KO animals.

Our RNAseq experiments in this paper did not actually look at HDAC7-KO animals but rather Vα14 x HDAC7-ΔP animals, however the point raised by the reviewer is still germane. We have acknowledged in the text (subsection “HDAC7 and PLZF Inversely Regulate a Shared Innate Effector Gene Network That is Highly Relevant to Autoimmune Disease”, first paragraph) that the different distribution of CD44/NK1.1 between the samples we compared could yield significant population effects based merely on differential representation of developmental subsets. However many of the important suppressive changes we see, e.g. suppression of Hobit, Icos, and ID2 in thymus (Figure 5A), are of a magnitude which exceeds what could be accounted for by this mechanism as they are actually expressed at lower levels than in conventional SP thymocytes. There are moreover many genes affected by HDAC7-ΔP in our analysis that are not normally changed during iNKT development (see Figure 5—figure supplement 1C), and conversely population effects alone cannot account for all the gene expression differences between HDAC7-ΔP X Vα14 thymocytes and CD4SP cells that were the same as those observed in the absence of HDAC7-ΔP (Figure 5A, diagonals). Lastly, many of the genes seen to be suppressed are normally already strongly upregulated at stage 1, which is abundantly represented in Vα14 x HDAC7-ΔP thymocytes (Figure 3—figure supplement 1D). Thus, while we acknowledge that this was not a perfect experiment we remain convinced that it does provide important information about the molecular-level effects of HDAC7 on iNKT differentiation.

Note: In the sections below, we provide new responses only to those reviewer comments not already addressed above.

Again, the full text of all reviews are given below. Please note that a revised manuscript should address all of the concerns cited in the full reviews, though the major issues are listed above and should be addressed separately in responses to the reviewers.Reviewer #1:[…] This is an interesting paper that clearly demonstrates an important role for HDAC7 in NKT cell development and function. My comments (below) are mostly technical and while they should be addressed and might slightly change the narrative of the manuscript, they will not change the overall message.In Figure 1 (as well as several other figures in the manuscript), the various gates used for the analysis by flow cytometry seem to be placed freely. For example, the gate used to define the CD1d tetramer+ cells often incorporates cells that are TCR negative (Figure 1, Figure 3B, Figure 4). It would have been useful to stain with the control CD1d tetramer to ensure that the gated events are real NKT cells. While this reviewer agrees that it does not change the overall results (that NKT cells are affected by HDAC7), it can change the interpretation of certain results. For example, very few events corresponding to NKT cells are left in the HDAC7ΔP mice. By gating these events, it is concluded that the cells are very immature (CD44-, CD24+, NK1.1-) and blocked in their development. It would be worthwhile to verify that these very few events are real NKT cells.Stage 0 NKT cells represent 300-1000 cells per thymus and are usually only visualized after magnetic bead enrichment of tetramer+ cells. These experiments were not carried out in that fashion here and the CD24+ cells would benefit from the examination of other markers to make sure that they represent the cells that the authors think they examine. To this end, stage 0 NKT cells are also CD69+ and Egr2+. Similarly, it is unclear to this reviewer why only 9% of the NKT cells in wildtype mice express NK1.1 (Figure 1C), while in B6 mice, the vast majority of these cells are in fact NK1.1+ (as seen in Figure 3C for example). Were all experiments performed with mice of the same age? Is it a parameter that matters?In Figure 2, it is unclear why Figure 2B shows staining in the spleen while the other panels of the figures pertain to the thymus. Similarly to the previous comment, it is unclear why only 30% of the cells would be expressing NK1.1 in that tissue.

We believe we have responded fully to all of these issues above.

The development of "innate" CD8 T cells that are CD44+, CD122+, Eomes+ was reported to be dependent upon IL-4 signaling in the thymus. The source of IL-4 in the thymus is thought to be NKT2 and PLZF+ gd T cells. The presented experiments clearly demonstrate that the increased proportion of CD8^+^ CD44+ Eomes+ in HDAC7ΔP mice is decoupled from an increased proportion of NKT2 cells. Did the author examine whether this was compensated by an increase proportion of gd PLZF+ T cells? (Figure 4 would argue that it is not although it was not formally tested).

We addressed the question of PLZF+ γδ T cells above.

The author should also modify the text of the manuscript in that the "innate" programs that might be regulated by HDAC7 are not the same in CD8 and NKT cells. NKT cells do not express Eomes and CD8 T cells do not express PLZF so the statement that "loss of HDAC7 results in the aberrant adoption of NKT-like innate effector programming" is not correct.

We acknowledge that we were comparing apples and oranges with this statement, and have therefore deleted the phrase “iNKT-like” from this sentence. However, since the abnormalities we observed in both CD4 and CD8 T cells from HDAC7-KO mice went beyond just EOMES expression and did at least superficially resemble the effects of transgenic PLZF expression (Figure 4E), we feel that retaining the rest of this statement was still appropriate.

NKT cell can adopt different fate in the thymus (NKT1, NKT2 and NKT17) that can be visualized by the expression of the various master transcription factors associated with the phenotype. It would have been interesting to stain NKT cells for Tbet, Rorgt in addition to PLZF.

We agree that this type of analysis would have added significantly to our understanding of the effect of HDAC7-ΔP on iNKT development, although the strong NKT1 bias of B6 mice (Lee, et al., 2013) perhaps does not make this the best strain background in which to ask such questions. We can however point to our gene expression analysis, in which GATA3, RORγ, and T-bet were all found to be downregulated in thymus and/or spleen (see Figure 5—figure supplement 1).

*The expression of PLZF protein is essentially gone in HDAC7ΔP mice (Figure 4A), yet analysis of mRNA expression (Figure 5A) reveals a very modest loss of Zbtb16 mRNA expression. How do the authors reconcile these findings? Does HDAC7 binding to PLZF (Figure 7) leads to its degradation* in vivo*?*

The source of this discrepancy is that our gene expression analysis was done in the Vα14/Jα18 transgenic background, in which PLZF expression, albeit reduced, is still detectable by flow cytometry in the presence of HDAC7-ΔP (Figure 4C).

It is also unclear why gene expression in total NKT cells was examined in Figure 5, while it is clear from Figure 1 that the cells did not reach the same stage of development between WT and HDAC7ΔP mice. Thereby, finding differences in gene expression for genes that are acquired late in the development of NKT cells (NK1.1, Ly49, Tbet) is perhaps not surprising, but might be unrelated to HDAC7 direct activity. It would have been more interesting, perhaps, to test differential gene expression in CD44neg NK1.1- NKT cells between the two strains.

We acknowledge that it might have been better to do the experiment in this fashion, and that for this reason some of the changes we see cannot be directly attributed to the effect of HDAC7. We have included a statement to this effect in the main text. However, as we detailed above, many of the important changes we did see occur earlier in iNKT development and/or are of a magnitude/direction that cannot be accounted for by population effects alone.

Reviewer #3:The main conclusion is that HDAC7 is critical for controlling whether T cells act like conventional naïve T cells or quickly reactive innate-type lymphocytes. However, there are several concerns and potential alternative interpretations.Data is not shown regarding the conventional CD4 compartment in the periphery. They state "We observe a much more modest degree of abnormality in the CD4 compartment (data not shown)". This data needs to be added to the Results and is necessary to assess their conclusions. If HDAC7ΔP does not alter the conv T cell compartment into a more innate like phenotype, does this alter the conclusions drawn?

We have now included these data for HDAC7-KO mice, as detailed above, as it was actually with this genotype and not HDAC7-ΔP that we saw these abnormalities.

Previously, the authors demonstrated that use of the HDAC7ΔP transgene inhibited negative selection of autoreactive thymocytes and that these mice developed lethal autoimmunity. Is the enhanced activation of T cells observed due to activation of autoreactive cells that otherwise would have been deleted? If these are autoreactive cells, then are they more innate like or the issue is that they are being activated and are thus not naïve? If the T cells with enhanced function are ones that would have been negatively selected, then is the primary cause the block in negative selection?

While we did in fact observe enhanced activation of autoreactive peripheral T cells in older HDAC7-ΔP TG mice our earlier work (Kasler, et al., 2012), this is not what we saw in iNKT cells from younger mice in the present work (see Figure 3C-D). The increased effector differentiation was rather observed in HDAC7-KO mice (Figure 2B-H, Figure 2—figure supplement 1B-C).

Another difficulty with this manuscript are conclusions 'consistent with the data', or 'suggesting that' without doing the experiments to demonstrate the mechanism. The authors interpret the failure of Va14 tg HDAC7ΔP NKT cells to produce cytokines as 'suggesting that the cells had failed to undergo effector programming in the thymus'. This should be examined by staining for NKT1/NKT2/NKT17 using Tbet/Gata3/RORγt. Total NKT are examined for cytokine production, but if there is a defect in differentiation of the NKT subsets, then each subset should be examined individually to determine whether (for example) there are few NKT1 cells that produce normal IFNγ or many NKT1 that fail to make IFNγ. In addition, the authors state that "HDAC7ΔP blocked the intrathymic proliferation that is normally associated with post-positive selection iNKT differentiation". No experiments were performed to examine NKT proliferation after positive selection (e.g. BrdU). In addition, the loss of NKT cell at this stage could also be due another reason such as enhanced apoptosis, but this was not examined either.

We have addressed these issues in responses made above, and have backed off of several statements, including the one quoted, that could be seen as overreaching.

"modest suppression of Treg and CD8aa IELs (data not shown)" should be added to the manuscript. CD8aa IELs are innate-like lymphocytes. If HDAC7ΔP expression leads to enhancement of innate functions, then why is this population decreased?

We have now added these data to the manuscript (Figure 1—figure supplement 1A), and must reiterate that we said there was enhanced effector function with loss of HDAC7, not with expression of HDAC7-ΔP.

It is not clear that the HDAC7ΔP transgene is completely off in the periphery. In their prior manuscript originally describing this mouse, there is enhanced HDAC7 expression in peripheral T cells from HDAC7ΔP transgenic mice as compared to WT. Therefore, I would argue that their previous paper demonstrated that this transgene is not completed turned off in the periphery.

While we acknowledge that the blot shown in that paper does show slightly higher expression of HDAC7 in splenocytes, what we assayed was total HDAC7 and not some epitope tag associated with the transgene. It cannot therefore automatically be concluded that this represents expression of the transgene rather than HDAC7. Even if it does, this could be due to recent thymic emigrants that have not yet fully turned off expression mediated by the lck proximal promoter.

HDAC7ΔP mice have increased mature SP thymocytes – one common cause for an increase in mature SP thymocytes is a defect in thymic egress. Is there a defect in thymic egress in these mice? (KLF2, S1P1/receptor, CD69)

Since HDAC7-ΔP-derived cells generally seem to contribute well to the peripheral Tconv population (e.g. Figure 1—figure supplement 1B), we did not think that investigating this possibility was critical to our conclusions, since any defect in thymic egress must not be very severe. The profound defect in negative selection we demonstrated previously (Kasler et al., 2012) also provides a pretty good explanation for the excess SP thymocytes we observe.

It is difficult to analyze the RNA-seq data presented, as the cells being examined are completely different: Vα14 (WT) NKT cells that have primarily differentiated into NKT1/NKT2/NKT17 effectors and Vα14 HDAC7ΔP NKTs that may not have differentiated into NKT1/NKT2/NKT17 effectors at all.

While we acknowledge that we did not do a perfect experiment here, we believe that showing as we did by gene expression analysis that such differentiation has not occurred is clearly a worthwhile end in itself – indeed while we use NK1.1 and CD44 for the staging of these cells, if we observed a defect in their upregulation in the absence of more comprehensive gene expression data, it might just as validly be argued that HDAC7-ΔP affects only the expression of those genes and not the whole program of innate effector differentiation.

Empty tetramer control needs to be added to several figures, as it is not clear whether the signal observed with Tetramer/TCRβ (e.g. Figure 1) in HDAC7ΔP is above that of an empty tetramer control. If not, then further examination of this population by CD24/CD44/NK1.1 may be misleading, and the population should be examined to determine whether it expresses the canonical Vα14-Jα18 rearranged TCRα chain.

As stated above, we have added empty tetramer to the key experiments, and have shown that we detect significantly more cells with loaded than with empty tetramer in HDAC7-ΔP thymus at all stages except Stage 1 (Figure 1—figure supplement 1C). We have also used magnetic enrichment, as suggested by reviewer 1, to increase confidence in our findings, obtaining a result very similar to what we saw without such enrichment (Figure 1C).